# Spatial and functional arrangement of Ebola virus polymerase inside phase-separated viral factories

Jingru Fang [1,2], Guillaume Castillon [3], Sebastien Phan[3], Sara McArdle [1], Chitra Hariharan[1], Aiyana Adams[1], Mark H. Ellisman [3], Ashok A. Deniz[2] & Erica Ollmann Saphire [1] ✉

Ebola virus (EBOV) infection induces the formation of membrane-less, cytoplasmic compartments termed viral factories, in which multiple viral proteins gather and coordinate viral transcription, replication, and assembly. Key to viral factory function is the recruitment of EBOV polymerase, a multifunctional machine that mediates transcription and replication of the viral RNA genome. We show that intracellularly reconstituted EBOV viral factories are biomolecular condensates, with composition-dependent internal exchange dynamics that likely facilitates viral replication. Within the viral factory, we found the EBOV polymerase clusters into foci. The distance between these foci increases when viral replication is enabled. In addition to the typical droplet-like viral factories, we report the formation of network-like viral factories during EBOV infection. Unlike droplet-like viral factories, network-like factories are inactive for EBOV nucleocapsid assembly. This unique view of EBOV propagation suggests a form-to-function relationship that describes how physical properties and internal structures of biomolecular condensates influence viral biogenesis.

Viruses are architects of cellular remodeling needed to fulfill life-cycle events inside host cells. Some viruses remodel host cells by inducing formation of membrane-less compartments, also termed viral factories (VFs), that separate essential viral replication and assembly events from other cellular processes[1–11]. As exemplified by many known cellular membrane-less organelles, such as the multiphasic nucleolus[12], different cellular processes can be spatially separated into distinct co-existing phases inside the membrane-less compartment. In contrast, how multiple interdependent viral biogenesis steps are coordinated inside viral factories remains unclear.

For viruses that induce VF formation, many are non-segmented, negative-strand RNA viruses (nNSVs)[2,3,5–8,10,11]. This group of viruses include important human pathogens such as rabies virus, measles virus, respiratory syncytial virus (RSV), and Ebola virus. Inside the virion, the RNA genome of nNSVs is coated by viral nucleoprotein to form a helical RNP structure. When the virus enters a host cell, the viral genome is released into the cytoplasm and the helical RNP relaxes, which allows the viral polymerase to access and transcribe the negative-strand RNA genome into multiple mRNAs encoding individual viral genes. Viral mRNAs are translated by host cell ribosomes into viral proteins. The same viral polymerase also coordinates synthesis of a complementary strand, known as the anti-genome, which templates the replication of progeny genomes. Genome and anti-genome are immediately wrapped with nucleoprotein oligomers to form RNP-helices. However, only the genome-containing RNP (vRNP), which is linked to viral polymerase, is assembled into progeny virions[13]. The synthesis of distinct viral RNA species and sorting the corresponding viral RNA species for

[1]La Jolla Institute for Immunology, La Jolla, CA, USA. [2]Scripps Research, La Jolla, CA, USA. [3]National Center for Microscopy and Imaging Research, Center for Research in Biological Systems, Department of Neurosciences, University of California San Diego, School of Medicine, La Jolla, CA, USA. ✉e-mail: erica@lji.org

translation vs. assembly all happen inside the cytoplasmic space, thus likely require some spatial regulation.

Recent studies indicated that several nNSV VFs are biomolecular condensates[11,14–17] with druggable material properties that correlate with virus inhibition[18]. Biomolecular condensate formation adheres to phase separation principles where molecules divide from a mixed system into phases with distinct constituents at varying concentrations in each phase[19,20]. This process is propelled by multivalent intermolecular interactions and impacted by the physiochemical and sequence-specific characteristics of proteins (including structured and disordered regions) and RNA components[21–24]. Under physiological conditions, the concentration of key constituents essentially controls the state of intracellular phase separation, which can be approximated in a phase diagram[19]. Deciphering phase behaviors of intracellular VF condensates and the spatial localization of distinct steps of virus biogenesis within them will outline the form-to-function relationship of VFs and enhance our understanding of nNSV replication.

Among nNSVs, Ebola virus is a zoonotic, human pathogen that causes near-annual outbreaks of disease with up to 90% mortality[25,26]. The only FDA-approved vaccine for Ebola virus protects against only Ebola virus Zaire (EBOV), and not the other pathogenic species. Further, therapeutic antibodies approved thus far are also specific for EBOV and may poorly penetrate immune-privileged sites where EBOV can lurk. Additional therapeutic strategies are needed, and will be accelerated by a better understanding of essential mechanisms in EBOV replication.

A hallmark of EBOV infection is the formation of eosinophilic, cytoplasmic inclusion bodies[27], which we now refer to as viral factories (VFs). EBOV nucleoprotein (NP) is considered the building block of VFs: in infected cells, VFs always contain NP, and in transfected cells, entropically expressed NP can induce VF-like, cytoplasmic granules. VFs recruit EBOV large polymerase protein L, polymerase cofactor VP35, and transcription factor VP30[28] through NP-VP35[29,30], L-VP35[31], and NP-VP30[32] interactions. Recruitment of L, VP35 and VP30 to EBOV VFs transform VFs into sites for active replication and transcription of viral RNA genome[6]. Further recruitment of EBOV minor matrix protein VP24 to VFs through NP-VP24 interaction[33] triggers the assembly of viral nucleocapsid. Given the biosafety level-4 requirement to work with live EBOV, reconstitution[29] and reverse genetic systems[33–35] have become routine practices to study molecular mechanisms underlining EBOV replication, transcription, and assembly or EBOV VFs formation. It is unclear whether the morphology, dynamics and internal forces of the VF condensates correlate with the coordination of EBOV biogenesis events.

Here, we define the spatial regulation of viral RNA synthesis and the EBOV polymerase within intracellular viral factories. We determine that reconstituted EBOV VFs exist as biomolecular condensates and dissect the contribution of key viral proteins to intracellular VF phase behaviors. Using multi-scale imaging approaches, we provide an unprecedented view of the spatial organization by which Ebola virus orchestrates multiple biogenesis steps and deploys viral replication machinery inside viral factories.

## Results

### An mNeonGreen-tagged EBOV VP35 can be used for live cell imaging of reconstituted viral factories (VFs)
Intracellular EBOV VFs are found to undergo fusion during infection[6]. This observation served as the first hint that EBOV VFs could be biomolecular condensates. Thus, we undertook a quantitative approach to determine whether EBOV VFs are indeed fluid condensates and define the internal molecular dynamics within EBOV VFs. Since microscopy of live, EBOV-infected cells under biosafety level-4 containment is neither feasible nor amenable to controlling VF composition, we first established a transfection-based, live-cell system to perform fluorescence recovery after photobleaching (FRAP) analysis with intracellularly reconstituted EBOV VFs.

To monitor EBOV VF with live-cell imaging, we adapted an approach used to image VFs in live cells infected with Rabies[15]. We fused an EBOV VF constituent, the polymerase cofactor VP35, to an N-terminal hemagglutinin (HA) epitope tag and a fluorescent protein, mNeonGreen (mNG) to generate HA-mNG-VP35. We chose to tag EBOV VP35 instead of another VF constituent, EBOV NP, because fluorescence-tagging of NP can ablate the biological function of NP in supporting viral RNA synthesis[36]. We confirmed that HA-mNG-VP35 expresses at a similar level as the wild-type (VP35-WT), and retains sufficient ability to act as the polymerase cofactor that supports EBOV viral RNA synthesis (Supplementary Fig. 1a).

VP35 interacts with EBOV nucleoprotein (NP) and chaperones the NP monomer ($NP^0$) prior to NP oligomerization[30]. As the polymerase cofactor, VP35 bridges the EBOV large polymerase protein (L) to NP-coated viral genome[28]. Since these multivalent intermolecular interactions VP35 involved in might provide the foundation for phase separation, we first performed co-immunoprecipitations and confirmed that the mNG tag on VP35 did not affect these interactions (Fig. 1a). Immunofluorescence microscopy showed that the intracellular localization of HA-mNG-VP35 resembles VP35-WT, when co-expressed with NP and L (Supplementary Fig. 1b). Together, our results indicate that HA-mNG-VP35 preserves sufficient polymerase-cofactor function and fully retains interactions with NP and L for use in live-cell imaging.

### Reconstituted EBOV VFs display composition-dependent, phase separation behaviors in live cells
To measure the internal exchange of EBOV VF components in a cellular context, we performed FRAP in HEK 293T cells co-transfected with EBOV NP and HA-mNG-VP35. Both NP and VP35 proteins contain intrinsically disordered and low complexity regions (Fig. 1b), bind RNA[37,38], and self-oligomerize[39–42], all common features associated with known cellular proteins to be involved in phase separation[19]. Here we used NP and VP35 as the minimal components to reconstitute EBOV VFs. Reconstituted EBOV VFs had diameters ranging from sub-micrometer to 10 μm (Supplementary Fig. 1c). Because the high mobility of sub-μm VF prevents accurate measurement and quantification of fluorescence recovery, we focused on medium-sized VFs (4–5 μm diameter) and photobleached a center spot within these sizable VFs (see "Methods").

As a control, HA-mNG-VP35 expressed alone phase separated in the cytoplasm (Fig. 1c). Most cells have large quantities of small (sub-μm) VP35-containing dense phases (condensates) with high apparent mobility relative to the dilute phase, but a small number of condensates had a larger size suitable for FRAP. These large condensates have multiple internal regions lacking fluorescence, which coincides with a "sponge-like" phenotype of VP35-condensates in a previous electron microscopy study[43]. Within these VP35-condensates, we observed slow internal molecular exchange (Fig. 1f, green curve), which could be mediated by VP35 self-oligomerization[41,42] or heterotypic interactions between VP35 and endogenous dsRNA[44,45], since diffusion of a similar-sized protein across a cytoplasmic spot (a circle with 2 μm diameter) occurs within milliseconds[46].

We next evaluated reconstituted EBOV VFs via co-expression of HA-mNG-VP35 and NP. The combination of HA-mNG-VP35 and NP yielded binary condensates (Fig. 1d) that appeared to be fluid-like based on frequently observed fusions between condensates or fission of one condensate into two parts (Supplementary Fig. 2a), and on non-fluorescence objects trafficking through VFs (Supplementary Fig. 2b). Over half the transfected cells had easily identifiable >5 μm condensates. Fluorescence recovery of VP35-NP condensates is more complete (mobile fraction ~85%) than that measured in VP35-alone condensate (mobile fraction <50%) (Fig. 1f, red vs. green curves). This result indicates that the VP35-NP interaction can mobilize molecules inside VF condensates. We also noticed the kinetics of internal

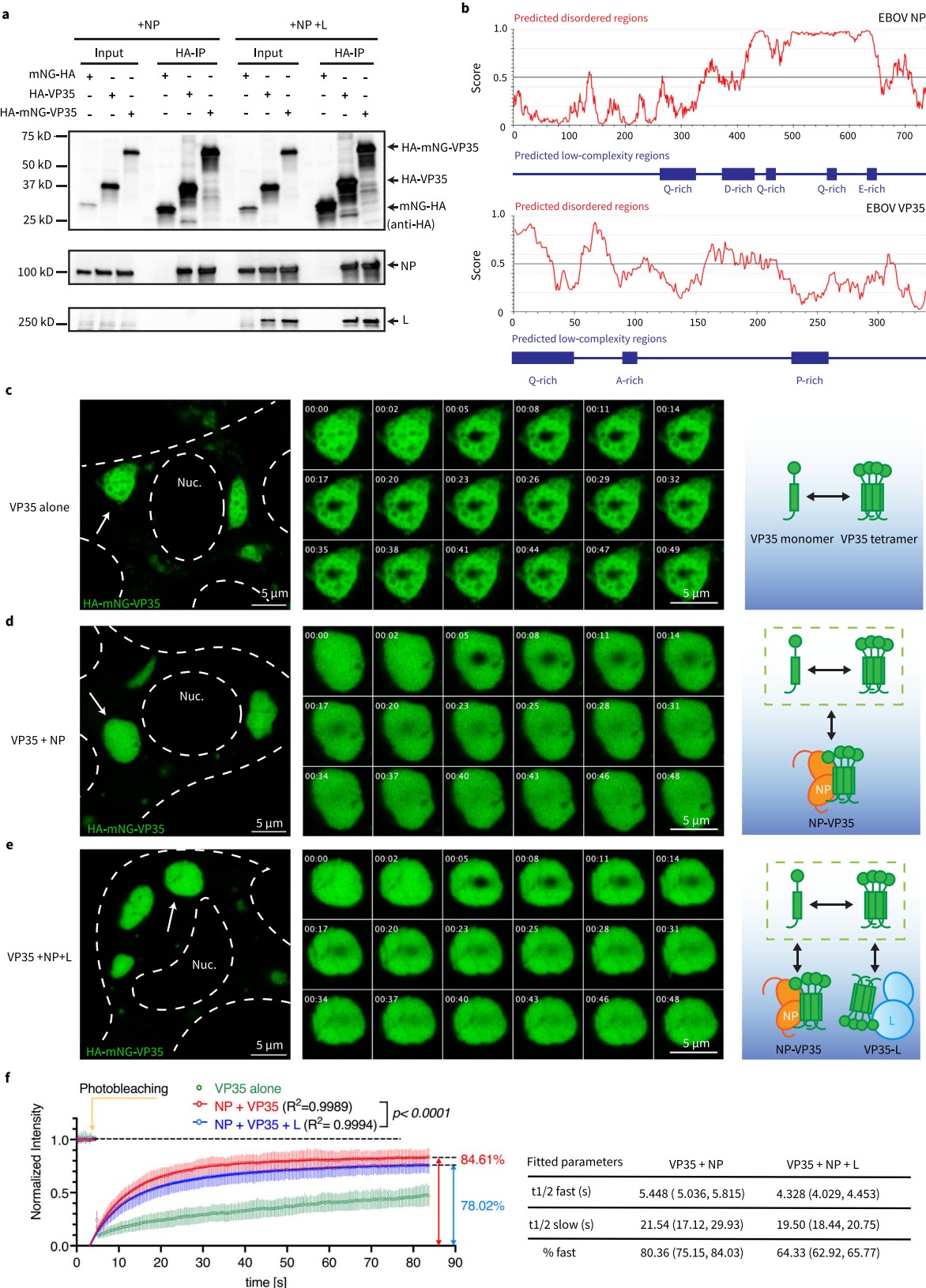

**f**

| Fitted parameters | VP35 + NP | VP35 + NP + L |
|---|---|---|
| t1/2 fast (s) | 5.448 ( 5.036, 5.815) | 4.328 (4.029, 4.453) |
| t1/2 slow (s) | 21.54 (17.12, 29.93) | 19.50 (18.44, 20.75) |
| % fast | 80.36 (75.15, 84.03) | 64.33 (62.92, 65.77) |

molecular exchange measured with Ebola VP35-NP condensates ($t_{1/2FAST}$ ~ 5.5 s) is similar with that reported for VFs in rabies virus infected cells[15] ($t_{1/2FAST}$ ~ 5.2 s) (Fig. 1f).

Co-expression of mNG-HA-VP35 with NP and EBOV large polymerase protein (L) resulted in dense phases in the cytoplasm that contained all three proteins (Fig. 1e, Supplementary Fig. 1b). In VP35-NP-L condensates, the mobile fraction of mNG-HA-VP35 (Fig. 1f, blue

curve) is on average 7% lower than VP35 + NP binary condensates and the percentage of molecules participating in fast association event (%fast) is on average 16% lower than binary condensates. This result suggested that VP35-L interactions could immobilize a small population of VP35. In both binary and ternary condensates, bleach pulses induced loss of fluorescence signal to around 12% of the prebleach signal at equilibrium (Supplementary Fig. 2c). Further, the similar

**Fig. 1 | Reconstituted EBOV viral factories with minimal viral protein components display viscoelastic behavior in live cells. a** Coimmunoprecipitation (co-IP) of EBOV NP and L with HA-mNG-VP35. HA-VP35 and HA-mNG served as positive and negative control, respectively. Anti-HA antibody was used to detect HA-mNG-VP35, HA-VP35, and HA-mNG in the input cell lysate and in HA-IP fraction. 1 mg of total proteins was used in each IP reaction. Representative results from 3 biological replicates are shown. **b** Bioinformatic prediction of intrinsically disordered regions and low complexity regions in the primary sequence of EBOV NP and V35 protein, using the IUPred and PlaToLoCo webserver, respectively. Single letter code denotes amino acid sequence. Confocal microscopy of **c** VP35, **d** VP35 + NP, **e** VP35 + NP + L condensates inside live HEK 293T cells 1 d post-transfection. Representative cells for each condition from 4 biological replicates are shown. The cell body and nucleus (Nuc.) are marked by a dashed line. White arrow: individual condensate chosen for photobleaching. Image montage is composed of selected frames (including $t = 0$ s) with an interval of 2.88 s from each time-lapse of photobleached condensate displayed. The diameter of photobleached regions is 1 μm.

Photobleaching occurred at $t = 4.8$ s. Scale bars: 5 μm. Schematic of EBOV VP35-involved molecular associations corresponding to each type of condensate. **f** Fluorescence recovery of mNG-HA-VP35 within the photobleached region inside intracellular condensates containing VP35. Normalized intensity corresponding to each time point before and after photobleaching of VP35 condensate is shown in green, VP35 + NP condensate is shown in red, VP35 + NP + L condensate is shown in blue. Each data point represents the mean with standard deviation (error bars) of $N = 6$ for VP35 condensates, $N = 9$ for VP35 + NP condensates, $N = 9$ for VP35 + NP + L condensates. Data points with $t > 5$ s in red and in blue were used to fit a corresponding two-phase association curve, with the normalized intensity value in expressed as a percentage and the curve plateau marked on the side. Goodness of fit of each curve is indicated by an $R^2$ value. An extra sum-of-squares F test (two-tailed) was performed to determine whether the best-fit value for unshared parameters differ between the blue and red curves. For each fitted curve, the best-fit value for each kinetic parameter is shown with a 95% confidence interval.

kinetics ($t_{1/2}$) for binary and ternary condensates likely indicates similar modes of molecular interactions are driving the VP35 mobility (Fig. 1f).

With the same composition of binary EBOV VFs, we next evaluated the contribution of an intrinsically disordered region (IDR) in EBOV NP to the internal exchange dynamics within VFs. Although EBOV VP35 also contains an IDR, which spans the first 80 residues on the N terminus (Fig. 1b), this region mediates the binding of VP35 to NP[0], and is thus too important to be removed for a deletion mutant[30]. In contrast, deletion of NP IDR (a.a. 500−739) partially preserves NP-VP35 interaction and recruitment of VP35 to NP-induced VFs[29]. We made a corresponding deletion mutant of NP, termed NPΔ (500−739), and co-expressed it with mNG-HA-VP35 (Supplementary Fig. 3a). However, the internal exchange dynamics of NPΔ (500−739)-VP35 binary condensates were similar to NP-VP35 condensates (Supplementary Fig. 3b, c), which means the IDR in EBOV NP is dispensable for controlling the internal exchange of binary VFs.

Our FRAP results quantitatively showed that EBOV VFs reconstituted with NP and VP35 display composition-dependent, internal exchange dynamics in live cells. The addition of L immobilizes a small, but detectable, fraction of VP35 inside the reconstituted VFs.

### FLAG-tagged EBOV large polymerase protein L forms foci inside EBOV VFs

We next assessed whether EBOV L is immobilized in a specific location within VFs using immunofluorescence microscopy. To facilitate detection of L, we engineered recombinant L with an N-terminal 2xFLAG tag (FLAG-L) as previously described[31] since the only currently available L polyclonal antibody can non-specifically bind to proteins other than EBOV L (Supplementary Fig. 1d).

First, we co-expressed NP, VP35, and FLAG-L in HEK 293T cells and labeled EBOV VFs using a monoclonal antibody targeting NP. Although NP was present throughout EBOV VFs, NP molecules at the VF periphery were immunolabeled more efficiently than the NP inside VF, leading to an "empty" droplet-like morphology of EBOV VFs, which was previously described[47,48]. We reasoned that this preferential immunolabeling of periphery NP could be due to the more expanded conformation of NP molecules and fewer intermolecular interactions located at the VF periphery than the interior[49]. Unexpectedly, FLAG-L was not homogenously distributed within reconstituted VFs and had a different staining pattern than NP (Fig. 2a). Instead, FLAG-L clustered into networks comprising interconnected foci within the ternary condensate, suggesting that an intrinsic property of L drives a different phase behavior than VP35 or NP.

To link this phase behavior of EBOV L to its biological function, we further analyzed L localization in the presence of an active RNA substrate for L, the EBOV minigenome (MG), which allows recapitulation of L-mediated replication and transcription. The EBOV MG we used here is bicistronic (2cis-MG), which contains two reporter genes encoding GFP (for imaging) and *Renilla* luciferase (for quantification) in a tandem cassette carrying authentic EBOV gene start- and end signals. EBOV regulatory sequences, the 3′ leader and 5′ trailer, required for replication, transcription, and encapsidation of viral RNAs[50], flank the bicistronic cassette. Using the replication-competent version of the 2cis-MG (Fig. 2b, Rep-comp. MG), we confirmed that the L construct, FLAG-L, we engineered is competent to fulfill its biological function as it retained 74% of the wildtype L (L-WT) activity (Fig. 2c).

In GFP-positive cells having successful reconstitution of EBOV MG replication and transcription, FLAG-L still clustered into networks of interconnected foci inside VFs (Fig. 2d). For comparison purposes, we chose not to consider the FLAG-L localization in GFP-negative cells, as expression of any EBOV protein or MG RNA might be lacking in these cells. Further, we noticed that due to chemical fixation, GFP reporter signals were preferentially trapped in both VFs and nucleoli in fixed cells.

We next asked whether modulating viral RNA synthesis affects the organization of EBOV L polymerase inside VFs. Since adding or removing MG system elements could affect the overall valency of protein-protein/RNA interactions or the molecular composition inside EBOV VFs, we incorporated a previously characterized mutation into the 5′ trailer of the EBOV 2cis-MG that allows transcription, but disables genome replication[51]. Reporter activity measured in this replication-deficient MG (Rep-def. MG) system reflects only viral transcription (Fig. 2b, Rep-def. MG). With Rep-def. MG, FLAG-L retained 33% of the L-WT activity, indicating that the 2xFLAG tag may have affected L-mediated transcription (Fig. 2c). Nevertheless, with FLAG-L, when replication was disabled, the L foci inside EBOV VFs were more closely spaced compared to that seen for replication-competent VFs (Fig. 2e, f). This different spacing is unlikely to be associated with the expression levels since FLAG-L was expressed at equivalent levels in the Rep-comp. and Rep-def. MG systems (Fig. 2c). Together, our results revealed a unique localization pattern of EBOV polymerase L inside VF and established a potential link between the spatial distribution of L within the VF and viral RNA synthesis events mediated by L.

### Network-like VFs exist in EBOV-GFP-ΔVP30 infected cells

Among the EBOV VFs reconstituted with the transcription and replication of EBOV MG in transfected HEK 293T cells, some display a granular, network-like morphology instead of the typical droplet-like morphology (Fig. 3a). We thus examined whether network-like VFs also occur during virus infection.

To characterize VF morphology in EBOV-infected cells, we used the biologically contained EBOV-GFP-ΔVP30 virus, which is morphologically indistinguishable from wild-type EBOV but approved for use in biosafety level 2+ containment[52]. In this system, a GFP gene replaces the gene encoding the viral transcription factor, VP30, so that EBOV-GFP-ΔVP30 virus can grow only in cell lines stably expressing VP30.

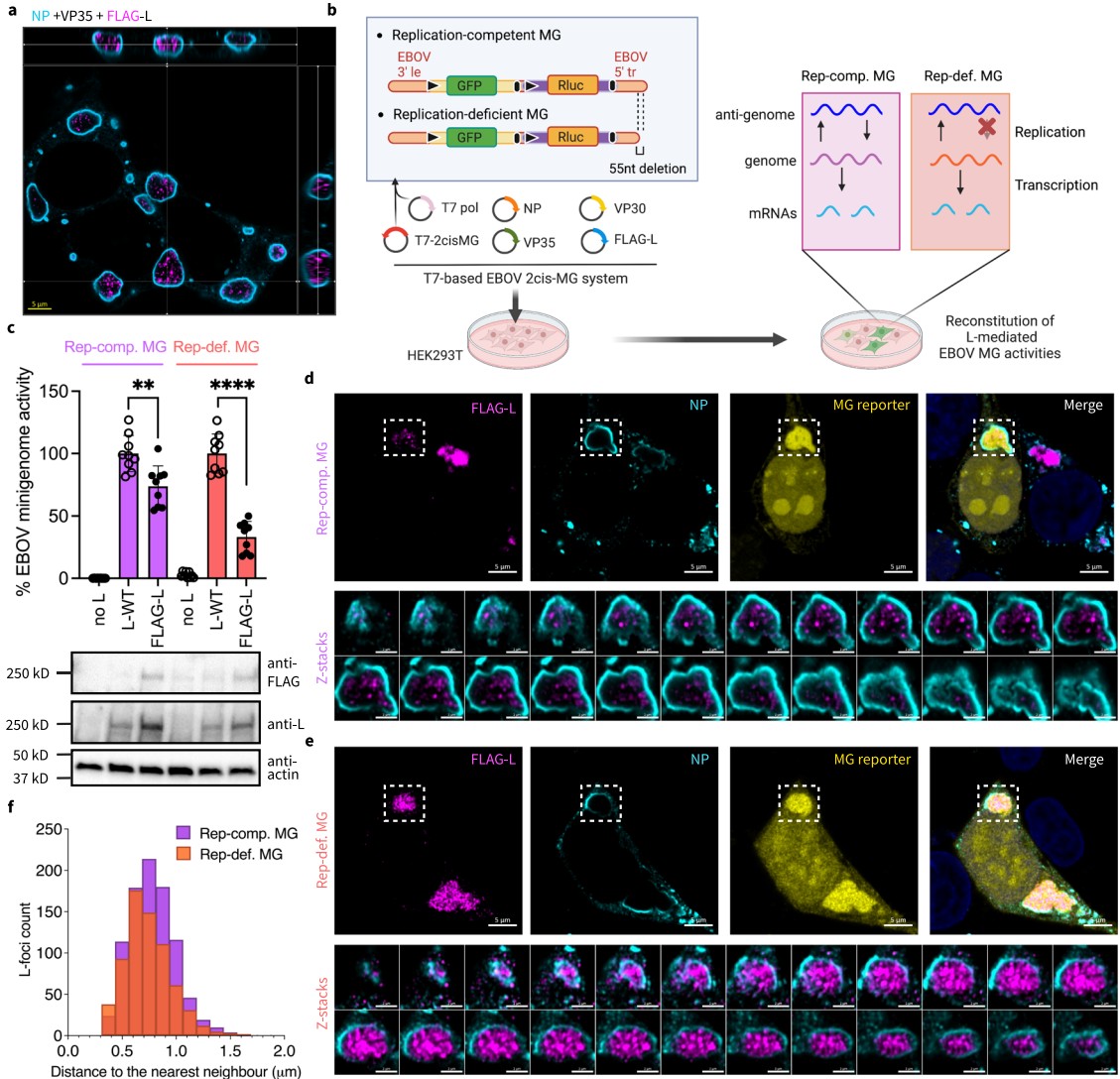

**Fig. 2 | FLAG-tagged EBOV L cluster into foci within reconstituted viral factories. a** Confocal immunofluorescence microscopy of fixed HEK 293T cells co-expressing EBOV NP, VP35, and FLAG-L at 1 d post-transfection. Represented Z-stacks in orthogonal view of 2 biological replicates each include >4 fields of view. Scale bars: 5 μm. **b** Schematics of the T7 polymerase (T7 pol), bicistronic EBOV minigenome (2cis-MG) system with a replication competent (Rep-comp.) or replication deficient (Rep-def.) MG. 3′ le: 3′ leader; 5′ tr: 5′ trailer; black triangle: gene start; black bar: gene end. Created with BioRender.com. **c** Activity of FLAG-L relative to the L-WT control measured in supporting the expression of *Renilla* luciferase (Rluc) in EBOV Rep-comp. vs. Rep-def. MG. Background expression of MG assessed by excluding L in the MG system (no L). Results from 3 biological replicates with technical triplicates are shown as individual data points with mean ± SD (error bars). **$p$ = 0.0024, ****$p$ < 0.0001 ($N$ = 9, two-tailed, unpaired t tests with Welch's correction). Expression of FLAG-L compared to L-WT and detection of the FLAG tag analyzed by western blot using a mouse monoclonal ani-FLAG and a rabbit poly-clonal anti-EBOV L antibody, respectively. Loading control: β-actin. Confocal microscopy of fixed HEK 293T cells transfected with **d** Rep-comp. or **e** Rep-def. EBOV MG system at 2 d post-transfection. A representative confocal image over-view is shown, in which selected EBOV VFs are marked with a white box (scale bars: 5 μm) and magnified in confocal z-stacks (scale bars: 2 μm). Fluorescence of the GFP reporter in both Rep-comp. and Rep-def. EBOV MG is pseudo-colored in yellow for display purposes. Nuclei are counterstained with Hoechst. Representative results from 4 biological replicates with >5 fields of view are shown. **f** A histogram showing the distribution of nearest distances between EBOV L-foci within the same VF in the presence of Rep-comp. vs. Rep-def MG, corresponding to z-stacks shown in (**d**) and (**e**). A total of 908 (Rep-comp.) or 683 (Rep-def.) L-foci used in quantification.

Here, we infected Vero cells stably expressing EBOV VP30 (Vero-VP30) with EBOV-GFP-ΔVP30 in a synchronized manner, and we fixed and inactivated these cells at 18 h post-infection. We chose this time point because levels of viral proteins and GFP reporters are above the detection threshold, and the VF size is comparable to that in HEK 293T cells transfected with the EBOV MG system[53]. We then used immunofluorescence-labeled NP as a marker to detect VFs in GFP-positive (i.e., EBOV-GFP-ΔVP30 infected) Vero-VP30 cells.

EBOV-GFP-ΔVP30 infected cells had either droplet-like or network-like VFs (Fig. 3b), with the majority (77%) having droplet-like VFs and a stronger immunofluorescence staining of NP at the VF periphery, as previously reported[53]. The remaining 23% of infected cells harbored network-like VFs, which, in contrast to droplet-like VFs located at discrete sites in the cytoplasm, occupied a more extended region that included a group of small, granular VFs either interconnected or separated by only a tiny distance. This network-like phenotype could reflect a later stage of EBOV life cycle when helical viral RNPs are assembled into filamentous nucleocapsids about 1 μm in length[54,55] and depart from VFs (Supplementary Fig. 4). The departure of assembled viral nucleocapsids can disintegrate VFs. However, we did not identify any assembled nucleocapsids nearby these network-like VFs (Fig. 3c–e), suggesting that these network-like VFs are not the immediate source of EBOV nucleo-capsid assembly.

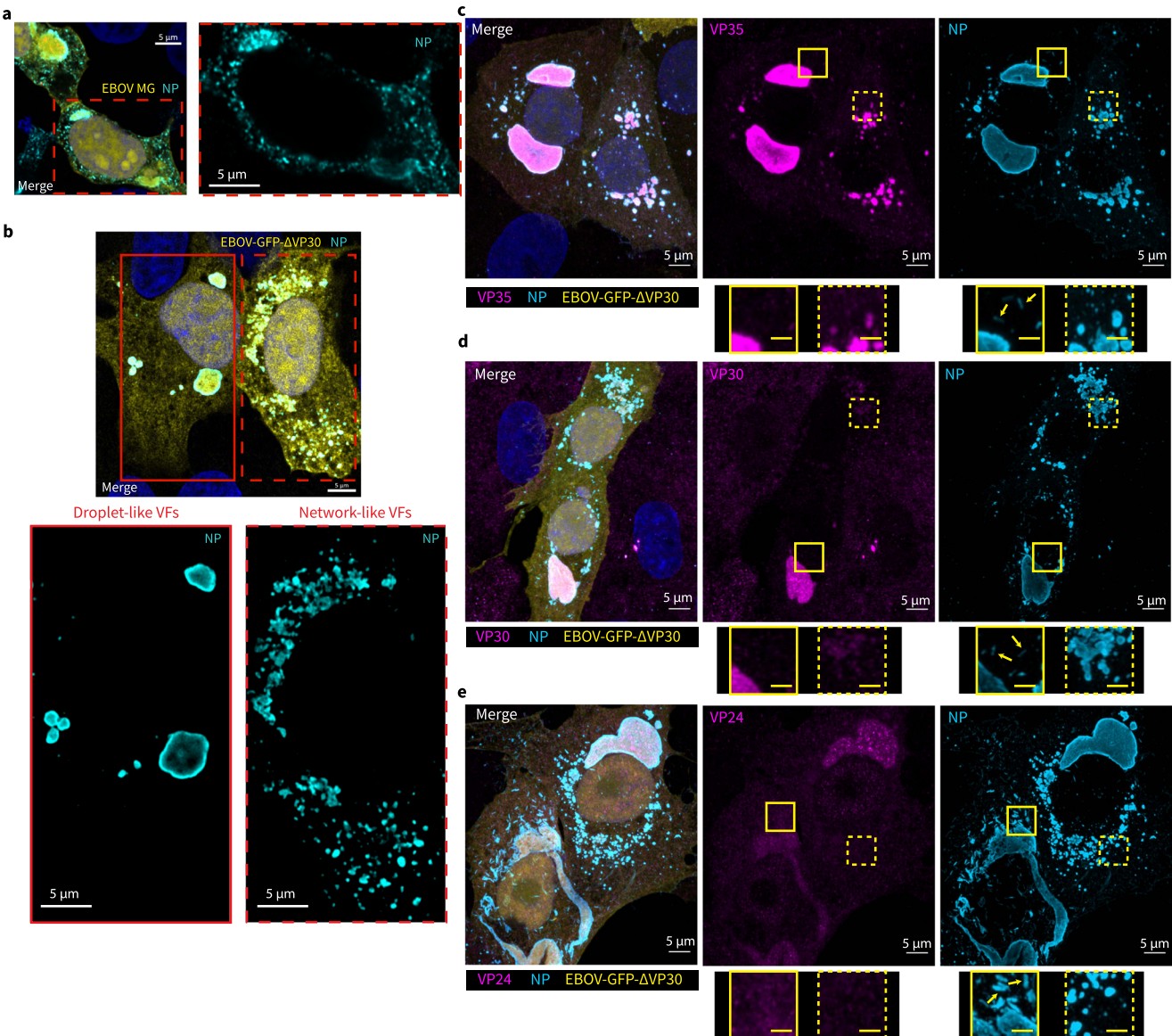

**Fig. 3 | Morphologically distinct EBOV viral factories in EBOV minigenome transfected cells and EBOV-GFP-ΔVP30 infected Vero-VP30 cells. a** Confocal immunofluorescence microscopy of HEK 293T cells transfected with T7-pol based, Rep-comp. EBOV MG system, at 2 d post transfection. L-WT was used. A representative result from 2 biological replicates with 3 fields of view is shown.
**b** Confocal immunofluorescence microscopy of representative Vero-VP30 cells infected with EBOV-GFP-ΔVP30 at MOI (multiplicity of infection) = 3, 18 h post infection. A total of N = 39 cells from 2 biological replicates were analyzed. EBOV NP was labeled with a human monoclonal anti-NP antibody paired with Alexa-568 anti-human antibody. A cell containing droplet-like viral factories is magnified in single-channel view with solid red outline. A cell containing network-like viral factories in dashed red outline. **c–e** Presence of EBOV nucleocapsid components in network-like viral factories was determined by double-immunofluorescence analysis in virus-infected Vero-VP30 cells with similar conditions as (**b**). In parallel samples, immu-nolabeled (**c**) EBOV VP35, (**d**) VP30, (**e**) VP24 were paired with Alexa-647 secondary antibody and analyzed in conjunction with NP immunofluorescence staining. Nucleus counterstained with Hoechst. Scale bars in white: 5 μm. Scale bars in yellow: 2 μm. Yellow arrows: viral nucleocapsid. Orthogonal projection of confocal z-stacks shown in (**c**), (**d**), and (**e**). Subcellular regions containing droplet-like viral factories are magnified in a single-channel view with solid yellow outline; regions containing network-like viral factories are shown with a dashed yellow outline. A total of N = 52 cells from 2 biological replicates were analyzed. For display purposes, fluorescence signals of the GFP reporter in Rep-comp. MG and in EBOV-GFP-ΔVP30 is pseudo-colored yellow; the brightness of insets in VP30 channel (**d**) and VP24 channel (**e**) are equivalently enhanced.

To understand why network-like VFs cannot support EBOV nucleocapsid assembly, we analyzed whether building blocks of viral nucleocapsids[56], which include EBOV NP, VP35, VP30, and VP24, are present in network-like VFs. We found network-like VFs contain EBOV NP and VP35 (Fig. 3c), a low level of VP30 (Fig. 3d), but not VP24 (Fig. 3e), which is necessary for nucleocapsid assembly[57]. Our results indicated that both the droplet-like and network-like morphology of EBOV VFs exist during EBOV infection. Given the expression of the GFP reporter, which is encoded in the viral genome, both types of VFs

support EBOV gene expression, but only the droplet-like VFs support viral nucleocapsid assembly.

## Engineering a split-APEX2 tag for electron microscopy analyses of EBOV polymerase L-VP35 complexes
To dissect the spatial organization of EBOV polymerase inside VFs at nanometer resolution, we used APEX2, a peroxidase tag engineered to indicate the location of tagged proteins in electron microscopy (EM) imaging. Specifically, we used the split-APEX2 (sAPEX) tag[58],

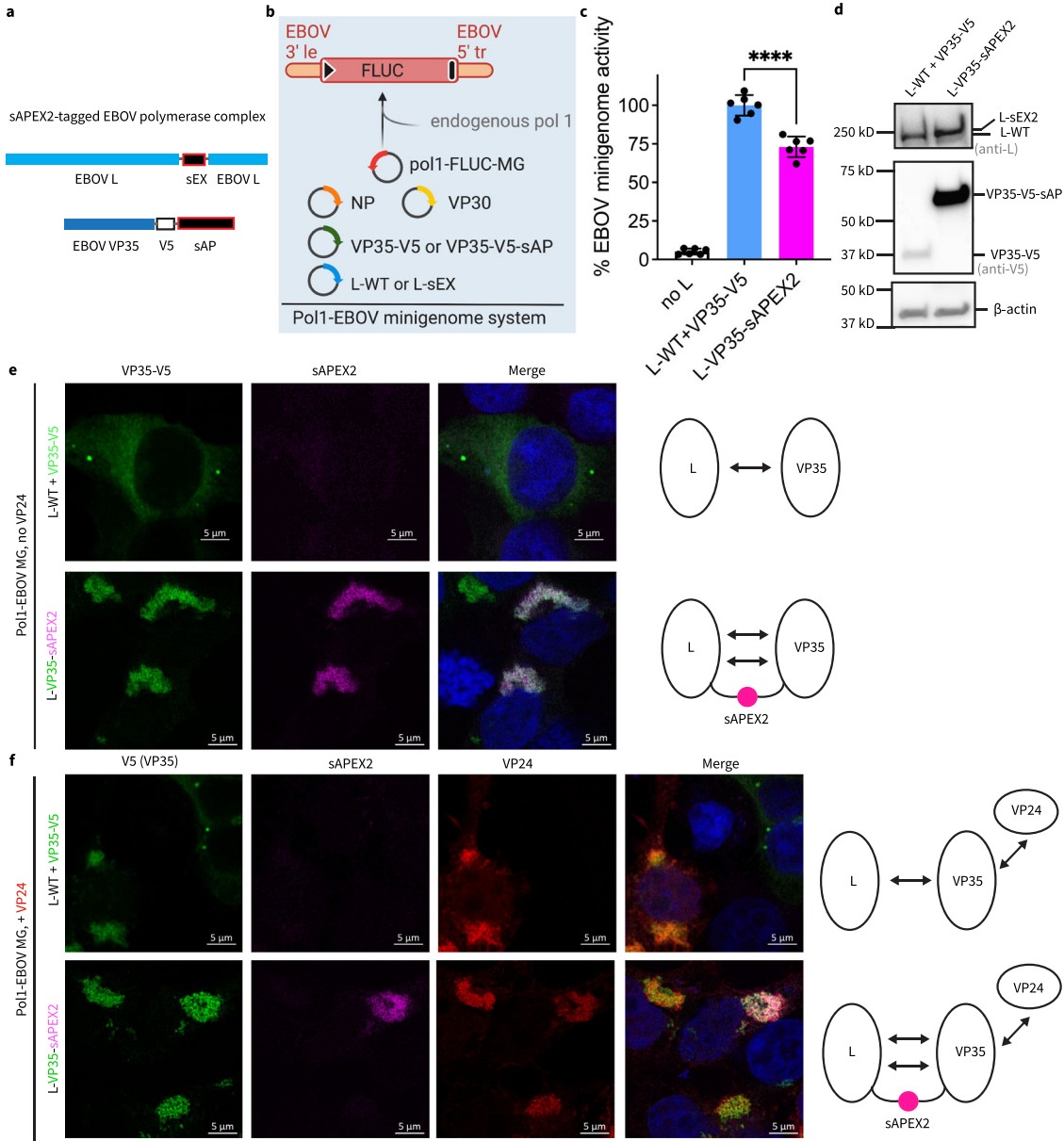

**Fig. 4 | Engineering and characterization of a split-APEX2 tagged EBOV polymerase complex. a** Protein constructs of the split-APEX2 (sAPEX2)-tagged EBOV L and VP35. sAPEX contains two fragments: sAP and sEX. **b** Schematic of the RNA polymerase 1(Pol1)-based, monocistronic EBOV minigenome system (Pol1-EBOV MG). 3′ le: 3′ leader; 5′ tr: 5′ trailer; black triangle: gene start; black bar: gene end. Created with BioRender.com. **c** Activity of L-sEX and VP35-V5-sEX (L-VP35-sAPEX2) relative to wild-type EBOV L (L-WT) and V5-tagged VP35 (VP35-V5) control measured in supporting expression of firefly luciferase (FLUC) in the EBOV MG. Background expression of MG assessed by excluding L in the MG system (no L). Results from 2 biological replicates with technical triplicates are shown as individual data points with mean ± SD (error bars). ****$p < 0.0001$ ($N = 6$, two-tailed, unpaired t tests with Welch's correction). **d** Expression of L-sEX compared to L-WT and VP35-V5-sAP compared to VP35-V5, each analyzed by western blot using a rabbit polyclonal anti-EBOV L antibody and a mouse monoclonal anti-V5 antibody, respectively. Loading control: β-actin. **e** Confocal immunofluorescence microscopy of HEK 293T cells

transfected with the Pol1-EBOV MG system containing either L-WT + VP35-V5 or L-VP35-sAPEX2, at 2 d post transfection. **f** Confocal immunofluorescence microscopy of HEK 293 T cells transfected with the Pol1-EBOV MG system containing either L-WT + VP35-V5 or L-VP35-sAPEX2, and in the presence of VP24, at 2 d post transfection. In (**e**) and (**f**), both VP35-V5-sAP and VP35-V5 were labeled with a rabbit monoclonal anti-V5 antibody paired with Alexa-647 anti-rabbit antibody. Functional reconstitution of APEX2 was indicated by fluorescence signals from resorufin, the product of the Amplex UltraRed, a fluorogenic peroxidase substrate for APEX2. In (**f**), EBOV VP24 was labeled with a mouse monoclonal anti-VP24 antibody paired with Alexa-488 anti-mouse antibody. Nuclei were counterstained with Hoechst. Fluorescence signals of VP35 and VP24 are pseudo-colored green and red for display purposes. Scale bars: 5 μm. Representative results from 2 biological replicates with >5 fields of view are shown. The valence of inter-molecular interaction is depicted as double-ended arrows.

consisting of two inactive fragments, sAP and sEX. We genetically fused the small fragment sEX to L (L-sEX) and the large fragment sAP, along with a V5 epitope tag, to VP35 (VP35-V5-sAP) (Fig. 4a). Interaction of L with VP35 during the formation of an active EBOV polymerase complex joins the sAP and sEX fragments to reconstitute APEX2 peroxidase activity.

We confirmed that sAPEX-tagged EBOV polymerase is functionally active, using a Pol1-based, monocistronic EBOV minigenome (Pol1-MG) system[35] that works similarly to the T7-based, bicistronic EBOV MG, but contains a single, firefly luciferase reporter gene (Fig. 4b, c). VP35-V5-sAP expression levels were significantly higher than the V5-tagged VP35 control (VP35-V5). L-sEX was also expressed to higher levels than

wild type L (Fig. 4d), suggesting the additional interaction provided by sAPEX could stabilize both L and VP35 proteins. In cells transfected with sAPEX2-tagged EBOV polymerase and the Pol1-MG system, we confirmed a site-specific reconstitution of sAPEX2 activity, as evidenced by APEX2-mediated conversion of a fluorogenic substrate at sites marked with VP35-V5-sAP immunofluorescence.

We noted that VP35-V5-sAP formed a network-like dense phase, whereas VP35-V5 formed droplet-like dense phases (Fig. 4e, L-WT + VP35-V5). These two morphologies of VP35-dense phase in transfected HEK293T cells are similar to those we observed with EBOV VF in infected cells, although network-like VFs in transfected cells showed higher connectivity than in virus-infected cells. We reasoned that network-like morphology of VP35-V5-sAP could be related to increased valence of the inter-molecular interaction in the sAPEX2-tagged L-VP35 complex, since trans-complementation of the sAPEX-tag creates an additional intermolecular interaction between L and VP35 (Fig. 4e, L-VP35-sAPEX2). To test this possibility independently of sAPEX2-tagging, we increased the valence of inter-molecular interactions that involve VP35 by adding VP24, an EBOV protein which also interacts with VP35[33]. Upon co-expression of VP24 with VP35-V5 and the Pol1-MG system, VP35-V5 droplets were replaced by the highly-connective, network-like VP35-V5 dense phase that colocalized with VP24 (Fig. 4f, L-WT + VP35-V5). This result indicates that increased valence of inter-molecular interactions indeed alters VP35 phase behavior. Even in the presence of VP24, the sAPEX2-tagged EBOV L-VP35 complex remained in the network-like dense phase (Fig. 4f, L-VP35-sAPEX2).

In summary, we successfully engineered an intracellularly active EBOV polymerase carrying a split-APEX2 tag that will allow localization of EBOV polymerase in electron microscopy (EM) analyses. Although the split-APEX2 tag did change the intracellular localization pattern of VP35 from droplets to networks, we have observed similar network-like VFs in EBOV-GFP-ΔVP30 infected cells. Therefore, we next carried out EM analysis on the localization of sAPEX2-tagged EBOV polymerase in cells reconstituted with the EBOV MG system.

## Nanoscale localization of split-APEX2-tagged EBOV polymerase complex with thin-section transmission electron microscopy (thin-section TEM)

We used the sAPEX2 tag engineered into the L-VP35 complex to locate the EBOV polymerase within the compact cellular contents revealed by EM. Upon staining with 3,3′-diaminobenzidine (DAB), trans-complementation of sAPEX2 catalyzes DAB polymerization with minimal diffusion. The resulting DAB polymers alone at the site of active APEX2 are chromogenic, which can be detected by light microscopy. Further, DAB polymers are osmiophilic, and thus capture osmium upon $OsO_4$ staining to increase electron density associated with DAB deposits (Fig. 5a). $OsO_4$ also stains unsaturated lipids[59] and reacts with nucleic acids[60,61] to outline cellular architecture in a specimen.

After optimization of transfection and staining conditions, we could detect sAPEX2-specific DAB deposits in bright field light microscopy images of HEK 293T cells transfected with sAPEX2-tagged EBOV polymerase together with other Pol1-MG components. DAB darkening was intensified after osmification and DAB-positive (DAB+) cells were readily recognizable in resin-embedded samples under light microscopy, which facilitates production of 70 nm thin-sections containing a DAB+ cell and its detection with transmission electron microscopy (TEM) (Fig. 5b). We observed within the same DAB+ cell network-like, electron-dense regions in cytoplasmic areas that likely correspond to EBOV VFs (Fig. 5c, Supplementary Fig. 5a). These electron-dense regions were present in cells transfected with the Pol1-MG system but not in untransfected control cells (Supplementary Fig. 5b, c). In the DAB+ cell, EBOV-specific electron-dense regions were decorated with darker dots arranged in a pattern that correlated with the shape of DAB darkening in light microscopy (Fig. 5c, arrows), suggesting that the

locations of these darker dots in the electron micrograph corresponded to sites where sAPEX2-tagged EBOV polymerases localize.

In the same thin-section, we also identified cells that had no DAB staining under light microscopy (DAB-) (Fig. 5b). These DAB- cells had the same transfection conditions as DAB+ cells, but sAPEX2 activity was not reconstituted. DAB- cells exhibited network-like, electron-dense VFs in the cytoplasm, similar to DAB+ cells (Fig. 5d). By comparing the morphology of EBOV VF in DAB+ and DAB- cells, the darker dots that associate with electron-dense VF in the DAB+ cell in fact indicate locations of sAPEX2-tagged EBOV polymerase (Fig. 5e).

Our thin-section TEM observation revealed that sAPEX2-tagged EBOV polymerase non-uniformly localizes to the periphery of network-like VFs. In several sites in the cytoplasm, sAPEX2-tagged EBOV polymerase preferentially clusters at the junction of interconnected VFs. Surrounding the electron-dense VFs were wire-like fragments that frequently associated with neighboring VFs (Fig. 5c, d).

## sAPEX2-tagged EBOV polymerase complex locate at the periphery or interconnected spots of network-like VFs

To visualize the three-dimensional (3D) ultrastructure of EBOV VFs and elucidate the spatial distribution of the sAPEX2-tagged polymerase complex, we applied electron tomography (ET) and reconstructed tomograms of subcellular volumes containing EBOV VFs and sAPEX2-tagged polymerase. Thin-section TEM allowed visualization of averaged 2D projection of all cellular content across the thin-section specimen, but multi-tilt electron tomography allowed us to resolve individual tomographic slices of the reconstructed volume with finer details of cellular matter (Fig. 6a, Supplementary Movie 1). Similar to previous EM analysis of virus-infected cells[3,62], reconstituted EBOV VFs appeared as electron-dense clusters that lack membrane boundaries themselves, but are in close proximity to membrane-bound cellular organelles including mitochondria, the endoplasmic reticulum, the nuclear envelope, and vesicles.

Across the tomographic slices, darker-stained patches near the edge of several electron-dense VFs were apparent. By mapping these darker patches in the tomogram, we reconstructed the 3D footprints of sAPEX2-tagged EBOV polymerase on a group of network-like VFs that were in close proximity or even interconnected (Fig. 6b, Supplementary Movie 1). We then selected a region of interest from the whole tomogram, and performed a finer manual segmentation focusing on smaller objects surrounding the VFs (Fig. 6c). sAPEX2-tagged polymerase appeared to dock at certain spots on the VF periphery. Ribosomes were located outside, but adjacent to EBOV VFs. We observed several loosely coiled structures either emerging from or passing through neighboring VFs, which likely gave rise to the rough surface of the VFs and are the major component of VFs (Fig. 6d). We propose that these loosely coiled structures are the viral RNP (vRNP), which has at its core helical viral genomic RNA bound to NP[55,63]. The helical structure is sufficiently relaxed to allow the viral polymerase to slide along the genome during RNA synthesis. We saw no condensed vRNP in our sample, likely because VP24, which is essential for condensing vRNP, was not included in the current MG system[64,65].

In a selected region from the whole tomogram, two loose-coil structures can, in fact, be traced back from their different branching sites to one parental loose coil (Fig. 6e). One possible explanation for these interconnected coils is that viral genomic RNA associates with active transcription products. For several better-resolved loose coils in our tomogram, the coil terminus often had a small loop linked to a solid globular structure. The identity of this terminal loop and globular structure remains to be determined. In another selected region, several sAPEX2-tagged EBOV spots were identified by manually tracing the darker stained patches on an invaginated edge of electron-dense VFs. These darker-stained patches localized to the apical sides of neighboring VFs where they formed a zipper-like structure (Fig. 6f). In reconstructed tomograms, the only obvious cellular structures that

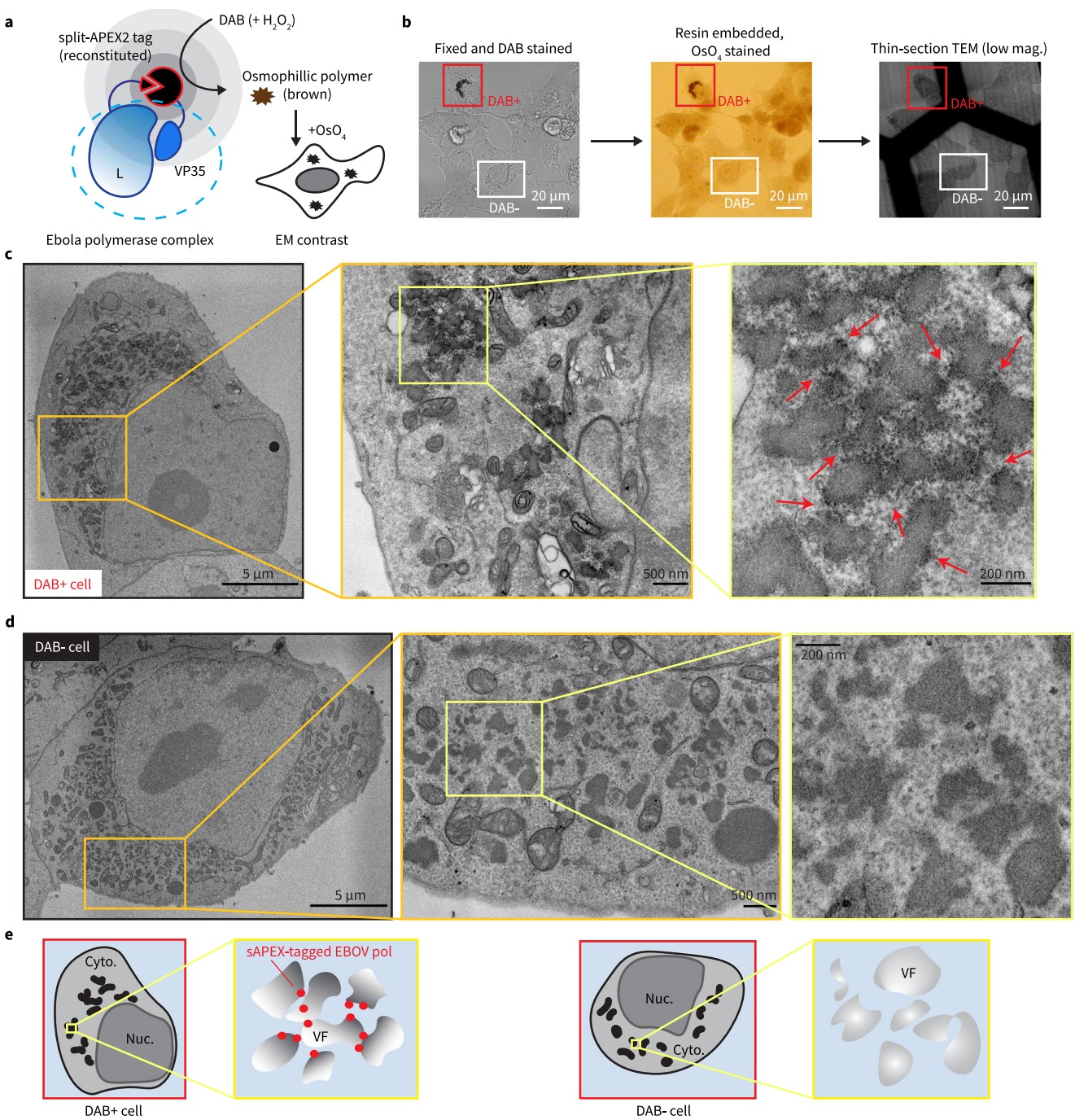

**Fig. 5 | Nanoscale localization of split-APEX2 tagged EBOV polymerase complex revealed by thin-section electron microscopy (EM). a** Schematic of the functional reconstitution of APEX2 upon formation of EBOV polymerase (L-VP35) complex and the resulting EM contrast upon DAB-OsO4 staining. **b** HEK 293T cells were transfected with the Pol1-EBOV-MG system containing sAPEX2-tagged EBOV polymerase (L-VP35) complex and were chemically fixed with 2.5% glutaraldehyde and stained with DAB at 2 d post-transfection. DAB+ and DAB- cells were examined by transmitted light microscopy (left) and then embedded in resin and stained with OsO4 (middle). The samples were serially sliced into 70 nm-thick sections and imaged with transmission electron microscopy (TEM) at low magnification (right).

Scale bars: 20 μm. **c** Electron micrographs of either the overview or a cytoplasmic region of the DAB+ cell indicated in (**b**) at different magnifications. Arrows: sAPEX2 mediated deposition of electron-dense Osmium. **d** Electron micrographs of either the overview or a cytoplasmic region of the DAB- cell indicated in (**b**) at different magnifications. Scale bars: 5 μm/500 nm/200 nm in micrographs with a low/medium/high magnification. **e** Cartoons annotating the whole cell overview, the electron-dense viral factories (VF) and the sAPEX2-tagged EBOV polymerases (sAPEX2-pol) in both the magnified DAB+ and DAB- cell, shown in (**c**) and (**d**), respectively. Representative results from 3 biological replicates with multiple fields of view are shown. Cyto.: cytoplasm. Nuc.: nucleus.

have direct contact with EBOV VFs are the single-membrane vesicles (Fig. 6b, f).

## Discussion

Ebola virus (EBOV) induces formation of viral factories (VFs) in host cells that allow physical separation of viral biogenesis from other cellular processes. In this study, we sought to understand the mechanisms by which EBOV VFs spatially accommodate and control viral RNA synthesis, with a focus on the localization of the EBOV polymerase complexes within VFs.

The contribution of individual VF components to the internal molecular exchange within VFs was poorly characterized in previous

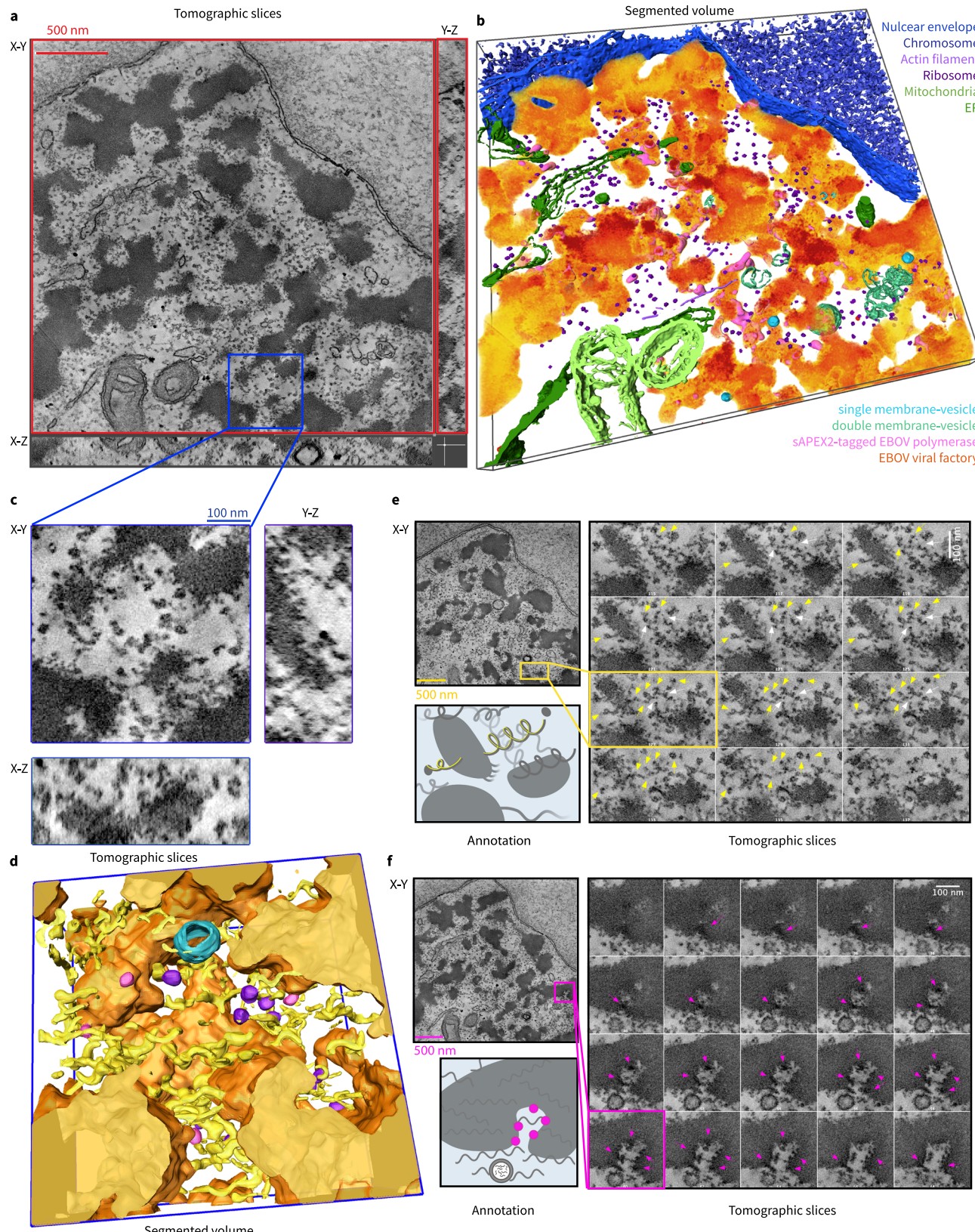

studies, even for other well-characterized viruses. Here, we used a transfection-based system to supply individual EBOV components for stepwise reconstitution of intracellular VF. The internal dynamics we measured using FRAP for EBOV VFs produced in transfected cells are strikingly comparable to those reported for VFs in rabies virus-infected cells[15]. Even though the stoichiometry of transfected viral proteins cannot fully recapitulate that of virus infection, our results demonstrate the validity of using transfection-based, intracellularly reconstituted VF to study phase separation behaviors of VF in living cells. Our key finding is a notable reduction in the molecular exchange rate within EBOV VFs upon recruitment of L to the NP-VP35 binary condensate, which is likely due to interaction of L with a fraction of mobile VP35. Reducing the

**Fig. 6 | 3D visualization of EBOV viral factories (VFs) and sAPEX2-tagged polymerase by electron tomography. a** HEK 293 cell transfected with the Pol1-EBOV-MG system containing sAPEX2-tagged EBOV polymerase (L-VP35) complex before fixation with 2.5% glutaraldehyde at 2 d post-transfection. The samples were stained with DAB-OsO$_4$, embedded in resin, and sectioned into 250-nm-thick specimens. A reconstructed 3D electron tomogram of the specimen originating from a DAB+ cell is shown in an orthogonal view with tomographic slices. Scale bar: 500 nm. **b** Segmented volume of the electron-dense EBOV VFs and surface representation of numerous cellular organelles and sAPEX2-tagged EBOV polymerase, corresponding to the 3D electron tomogram in (**a**). **c** Orthogonal view (with X-Y/Y-Z/X-Z plane) showing a magnified volume of interest in the 3D electron tomogram in (**a**). Scale bar: 100 nm. **d** Segmented surface representations of EBOV VFs (orange), sAPEX2-tagged EBOV polymerase (pink), viral RNP coils (yellow), host

ribosomes (purple), and single-membrane vesicle (blue), resulting from the tomographic volume in (**c**). **e** Series of magnified tomographic slices on the X-Y plane from a region containing multiple helical structures that correspond to viral RNPs. Yellow arrows pointing toward one helical structure; white arrows pointing toward the branch site where multiple helical structures meet. A cartoon annotation of the same region created with BioRender.com is shown on the side. **f** A series of magnified tomographic slices on the X-Y plane from a region containing DAB-osmium deposits that correspond to sAPEX-tagged EBOV polymerases (indicated by pink arrows) and a single-membrane vesicle approaching the electron-dense VF. A cartoon annotation of the same region created with BioRender.com is shown on the side. For (**e**) and (**f**), the scale bar in the tomographic overview slice and magnified series represents 500 nm and 100 nm, respectively. Tomographic slice number is marked in the bottom.

internal molecular exchange rate within EBOV VF could functionally impact viral RNA synthesis. Longer dwell times of EBOV polymerase and its cofactors inside VFs may facilitate the initiation step of EBOV polymerase-mediated viral RNA synthesis. A slower internal molecular exchange rate within EBOV VFs may also facilitate concurrent NP-coating of nascent RNA genome.

We observed that VP35 expressed in the absence of NP tends to form immobile aggregates, which indirectly supports a crucial role for NP as a modulator for the internal exchange of VF condensates. A previous study has shown that in the presence of VP35, a small part of the intrinsically disordered region (IDR) in NP (481–500) is critical but the rest of NP C-terminal region (500–739) is dispensable for VF formation. NP (481–500) also serves as a second binding site for NP-VP35 interaction and interacts with the well-folded VP35 C-terminal domain[29], in addition to the NP-VP35 interaction mediated by VP35 N-terminal IDR (20–48) binding to the well-folded NP N-terminal region (1–457) (Supplementary Fig. 3a). Here we extended this finding by asking whether the IDR in NP contributes to VFs internal exchange. We chose a NP mutant, NPΔ (500–739), of which the majority of IDR is removed but NP-VP35 interaction is retained, so that this mutant can still form condensates with VP35. Our results indicate that, in the presence of VP35, the majority of IDR in NP is also dispensable for regulating VFs internal exchange. It is tempting to conclude that the role of IDR in EBOV NP and VP35 in regulating VF phase behavior is coupled with high-affinity heterotypic interactions with modular protein domains, as revealed in previous structural studies[30,66]. This mode of interaction should be distinguished from weak interactions between IDRs that have been described in many other kinds of biomolecular condensates[20]. Future studies can determine the role that the oligomerization domains of EBOV NP and VP35 play in condensate formation and behavior. Our work lays out the experimental foundation to further understand the role of individual protein components and their domains in modulating VF material properties and corresponding functions.

Little was previously known about the functional localization of nNSV polymerases inside cells. The lack of specific antibodies suitable for immunofluorescence labeling and the low protein abundance of viral polymerase in infected cells has made imaging of nNSV polymerases particularly challenging. An early study first identified that mCherry-tagged EBOV L polymerase homogenously locates inside VFs during infection[6]. Despite the value of mCherry for live-cell imaging, the activity of L-mCherry was only 10% that of unmodified L and the integrity of the L-mCherry was unclear due to the lack of a detection antibody against L itself at the time of the study. In contrast to insertion of a larger fluorescence protein[67], L protein can tolerate a small epitope tag without substantial loss of RNA synthesis activity[36]. A recent study using super-resolution light microscopy examined the intracellular localization of FLAG-tagged RSV L in RSV-infected cells[68]. In that study, RSV L concentrated non-uniformly at several sites inside RSV VFs. This result aligns well with our confocal microscopy findings that FLAG-tagged EBOV L formed foci inside VFs in cells reconstituted

with the replication and expression of EBOV minigenome. A recombinant EBOV expressing FLAG-tagged L should be generated to further validate our findings with L localization in cells infected with authentic virus under biosafety level-4 containment.

Beyond elucidating the functional localization of EBOV polymerase L, our work links the localization pattern of EBOV L to particular types of L-mediated viral RNA synthesis. We hypothesized that VFs have distinct sub-compartments that are dedicated to viral replication or transcription. To understand EBOV L localization for replication or transcription, we separated the two processes by generating a replication-deficient EBOV minigenome (MG)[51]. We observed different EBOV L localizations in cells reconstituted with replication-competent vs. replication-deficient MG systems. When viral replication was active, the foci containing L were more widely spaced than when replication was disabled. A wider spacing of L-foci could allow more NP to permeate and become available for genome encapsidation[69]. Conversely, a narrower spacing of L-foci may reflect a switch from replication to transcription, which produces flexible viral mRNAs that are less bulky. Our result implies that rather than having distinct sub-compartments specialized for viral replication and transcription, the spatial organization of EBOV L expands and contracts according to the type or activity of ongoing viral RNA synthesis.

In addition to the spatial-functional dynamics with EBOV polymerase, we revealed an atypical, network-like morphology of EBOV VFs. In previous studies with virus-infected cells, nNSV VFs, including those induced by EBOV, appear mostly as droplet-like structures[6–8,70]. The droplet-like morphology of biomolecular condensates can be modeled by classical nucleation theory[71,72]. We did observe droplet-like VFs in EBOV-GFP-ΔVP30 infected cells, but a quarter of the cells contained network-like VFs, and sometimes both droplet-like and network-like VFs coexist in the same cell. Network-like VFs have not been reported for other viruses, except for Marburg virus[3], which is closely related to EBOV and also belongs to the *Filoviridae* family. Our research has shown that these network-like VFs are not involved in viral nucleocapsid assembly and that they lack the viral assembly factor, VP24. The absence of VP24 cannot be attributed to impaired viral gene expression since cells with network-like VFs can still express the GFP reporter encoded in the viral genome, nor to the composition of network-like VFs, because the presence of both EBOV NP and VP35 in these VFs should be sufficient to recruit VP24[33]. We believe that the presence of network-like virus factories (VFs) could indicate either (1) a stage in which the virus is transitioning from making viral mRNA and genomes to assembling the viral nucleocapsid or (2) a stage in which viral replication is not functioning properly due to dynamic arrest or asymmetry in phase separation[73,74]. To interrogate both hypotheses, one can follow the changes in the appearance of network-like VFs in live cells infected with fluorescence-tagged Ebola virus using light microscopy in a high-containment laboratory.

Finally, we visualized the precise 3D localization of EBOV polymerase with nanometer resolution in a subcellular volume using electron-tomography. Here EBOV polymerase with the split-APEX2 tag

allowed localization of the polymerase among the heterogenous intracellular content seen under electron microscopy[58]. Incorporating the split-APEX2 into L and VP35 preserved substantial levels of EBOV polymerase activity, although the valence of intermolecular interaction was increased. This increased valence of intermolecular interactions in VF components appeared to shift the EBOV VF morphology from the typical droplet-like to network-like structures, which echoes the network-like VF we observed in EBOV-GFP-ΔVP30 infected cells. Under such circumstances, we resolved multiple footprints of the EBOV polymerase gathering at discrete sites in network-like EBOV VFs, where the VF phase boundary was invaginated or interconnected. These sites may correlate to the EBOV L-foci under confocal light microscopy.

Based on our confocal light and electron microscopy findings, we propose a model in which EBOV polymerase molecules act in concert in a spatially restricted manner. Having a group of EBOV polymerase molecules acting together on the same viral genome would compensate for the low efficiency of viral RNA synthesis, as a productive viral RNA synthesis event can only initiate at a single site at the 3′ end of the viral genome[75]. Even so, during transcription, the EBOV polymerase makes frequent stops at each intergenic region, and at each stop the polymerase may detach from the genome template. How the same polymerase returns to the 3′ end of the viral genome and initiates a new round of RNA synthesis events is currently unclear. If a group of EBOV polymerases acts on the same viral genome, some polymerases will fall off the template at the gene stop signals, while other copies of the polymerase eventually reach the distal end of the genome where the L gene is located. By organizing the viral genome in tandem or in another coordinated way, the same group of polymerases could process more copies of the viral genome and maximize viral RNA production.

This study has limitations. First, our results with various kinds of cell-based reconstitution systems are only indicative, since these reconstitution systems cannot recapitulate the full EBOV life cycle and do not supply all viral proteins. Since we are using cell-based systems, cellular factors of the 293 and Vero cells used could inevitably influence VFs phase behavior and affect interpretation of our results. Second, we are aware of potential artifacts with chemical fixation[76], antibody-based immunofluorescence labeling, and split-APEX2 tagging of the viral polymerase for characterization of intracellular condensates. Live-cell imaging with fluorescence protein tagging may validate some of our findings. However, due to the bulkiness of fluorescence protein, incorporation of a fluorescence tag does come with a price (i.e., disruption of protein function) and may require high containment access for EBOV. The usage of tetracysteine tag in live cell imaging[77] should be explored for EBOV proteins in the future. Nevertheless, the results of our current study increase our understanding of the mechanisms associated with EBOV VFs and the spatial regulation of viral RNA synthesis within. Given the similar replication strategy shared among nNSVs, our findings may be applicable to other critically important nNSV pathogens, such as rabies, RSV and measles virus.

## Methods

### Plasmids

If not specified, NEB HiFi assembly was used for incorporation of DNA fragments into plasmid vector. To generate pCAGGS-mNG-HA-VP35, coding sequence of mNeonGreen (mNG) was PCR amplified from pmNeonGreenHO-3xFLAG-G (addgene#127914), while an N-terminal HA epitope tag was added to the mNG coding sequence. The resulting fragment was inserted upstream of the VP35 coding sequence in pCAGGS-VP35. To generate pCAGGS-HA-VP35 control plasmid, a DNA fragment containing a HA epitope tag followed by a flexible linker was synthesized and added to the N-terminus of VP35 coding frame. To generate pCMV-mNG-HA control plasmid, a HA epitope tag was added to the C-terminus of mNG coding sequence. To generate pCAGGS-

VP35-V5-sAP, the fragment of sAP was PCR amplified from FKBP_V5_AP_NEX_pLX304 (addgene#120912) and cloned into the intermediate vector pcDNA5-VP35. The fragment of VP35-V5-sAP was subcloned to pCAGGS plasmid using restriction sites Xho1 and Nhe1 and standard ligation method. To generate pCAGGS-VP35-V5 control plasmid, the fragment of VP35-V5 was PCR amplified with an addition of a stop codon and Bgl2 restriction site, and was then incorporated into pCAGGS backbone using Kpn1 and Bgl2 restriction sites.

To generate pCAGGS-FLAG-L, a DNA fragment containing a 2xFLAG epitope tag followed by a flexible linker was synthesized and added to the N-terminus of L coding frame. To generate pCEZ-L-sEX, a fragment of L coding sequence was first subcloned into a pFastbacDual intermediate plasmid using the natural restriction sites Pac1 and Hpa1. From there, coding sequence of sEX was PCR amplified from HA-Halotag-FRB-EX-NES-pLX304 (addgene #120913), flanked by two flexible linkers and was internally inserted to L fragment at the position 1705/1706 (TTIP/Q). The L-sEX fragment was shuffled back to the pCEZ-L backbone using the same restriction sites Pac1 and Hpa1.

To generate pT7-2cis-MG-Replication competent (Rep-Comp.), the coding sequence of VP40, GP, and VP24, including non-coding sequences VP40/GP and VP30/VP24 in between were removed from pT7-4cis-EBOV-vRNA-eGFP plasmid[34], while the coding sequence of *Renilla* luciferase was inserted downstream of the non-coding sequence NP/VP35 and reassembled. To generate pT7-2cis-MG-Replication deficient (Rep-def.), the last 55 nucleotide of the 5′ trailer sequence in pT7-2cis-MG-Rep-comp. plasmid was removed by PCR and re-assembled. Other supporting plasmids used in T7-based EBOV minigenome system were obtained from Dr. Thomas Hoenen (Friedrich-Loeffler-Institute). Plasmids used in Pol1-based EBOV minigenome system were obtained from Dr. Yoshihiro Kawaoka (University of Wisconsin Madison).

### Cell lines and virus

Human embryonic kidney cells HEK 293T (ATCC reference: CRL-3216) were originally purchased from the ATCC organization. Vero cells stabling expressing Ebola virus VP30 protein (Vero-VP30) were obtained from Dr. Yoshihiro Kawaoka (University of Wisconsin Madison). Both cell lines were maintained in Dulbecco's modified Eagle medium (DMEM-GlutaMAX) supplemented with 4.5 g/L D-Glucose, 10% fetal bovine serum (FBS), penicillin (100 U/mL), streptomycin (100 μg/mL). In addition, puromycin (5 μg/mL) was included in the culture media while maintaining Vero-VP30 cells but excluded for infection experiment. The biologically contained Ebola virus, EBOV-ΔVP30-GFP was generated in HEK 293T cells and grown in Vero-VP30 cells as previously described[52].

### Fluorescence recovery after photobleaching (FRAP)

IBIDI μ-slides (8 wells high-precision glass bottom) were treated with human fibronectin (50 mg/mL) for 20 min at 37 °C incubator, prior to HEK 293T cells seeding ($4 \times 10^4$ cells/well). Twenty-four hours later, the monolayer was transfected with plasmids in three different combinations: (1) 125 ng of pCAGGS-mNG-HA-VP35 and 1 μg of pCAGGS-vector control; (2) 125 ng of pCAGGS-mNG-HA-VP35, 125 ng of pCAGGS-NP, and 875 ng of pCAGGS-vector control; (3) 125 ng of pCAGGS-mNG-HA-VP35, 125 ng of pCAGGS-NP, and 875 ng of pCAGGS-L (wild-type). Transfected cells were incubated with complete FluoroBrite DMEM (+1x Glutamax, 10% FBS) for imaging. Time series were acquired using ZEISS laser scanning microscope (LSM880) with an Airyscan detector while live cells sample were kept in caged incubator maintaining 37 °C and 5% $CO_2$. mNG fluorescence was excited with a 488 nm argon laser and detected with a 63×1.4NA oil immersion objective (Plan-Apochromat), with a combination of bandpass filters (420–480) and (495–550). Using a frame rate of 240 ms per frame and 0.4% of 488 nm laser, 20 frames were recorded pre-bleach, and then 330 frames were recorded post-bleach under the Airyscan fast mode. This duration of

post-bleach image acquisition was experimentally determined by the time when fluorescence recovery reached a plateau. For photobleaching, 488 nm laser at 80% was used to pulse-bleach a spot size of 1 micron in diameter in the center of large condensates, until the fluorescence intensity of the bleach spot reduced to 30% of the prebleached value.

## FRAP quantification and interpretation

Time series in which the bleached object moved during post-bleach phase were manually identified and removed from data analysis because the fluorescence intensity of the bleached spot could not be accurately measured in ImageJ/FIJI[78]. For each bleach event, double normalization of the FRAP curve was performed using ImageJ/FIJI as described[15]. To correct unintentional photobleaching due to time-series acquisition, in every frame, fluorescence intensity of the bleached spot (1 μm in diameter) was first normalized to the mean intensity of an unbleached spot (1 μm in diameter) of another condensate in the same field of view. The resulting values after correction for all frames were further normalized to the average intensity of the last 10 pre-bleach frames. The background intensity in each field of view was negligible as it was lower than 0.5% of the average intensity of viral factory condensates.

Due to biological mobility of the intracellular condensate and the cellular specimens during imaging, we could not accurately measure the bleach pulses-induced fluorescence loss in the whole condensate. Instead, we assessed this parameter in the averaged fluorescence decay of internal reference-regions inside the same condensate that was photobleached but are away from the bleached spot. Double normalized data from 6 to 9 cells in a total of three independent experiments were pooled. Means with standard deviation of double normalized intensity were plotted as a function of time, of which post-bleach data-points were fitted to a two-phase association model in GraphPad prism, according to previously described methods[79]. Best-fit values with 95% confidence interval and goodness of fit indicated by $R^2$ for each data set were reported.

$$A_{fast} = (Plateau - Y_0) \times \%Fast \qquad (1)$$

$$A_{slow} = (Plateau - Y_0) \times (1 - \%Fast) \qquad (2)$$

$$Y(t) = Y_0 + A_{fast}\left(1 - e^{-k_{fast}t}\right) + A_{slow}(1 - e^{-k_{slow}t}) \qquad (3)$$

Fluorescence recovery of a sub-area within the VF condensate describes a combined rate of molecular exchange that occur within the VF condensate and occur between the VF and the cytoplasm. However, because the fluorescence intensity of molecules in the VF condensate is ~200-folds higher than that of the diffused molecules in the cytoplasm and the total VF condensate is significantly larger than the photo-bleached spot. The combined rate measured in the current work could approximate the internal exchange rate within the VF condensate.

## Immunofluorescence staining and confocal light microscopy

Fixation and staining was performed as described[80].

For samples shown in Supplementary Fig. S1b, human monoclonal anti-Ebola virus NP (1:2000), mouse monoclonal anti-Ebola virus VP35 (1:500), and rabbit polyclonal anti-Ebola virus L antibody (1:200) were used as the primary antibodies, goat anti-human Alexa-568 (1:1000), goat anti-mouse Alexa-647 (1:500), and goat anti-rabbit Alexa-568 (1:500) were used as the secondary antibodies.

For samples shown in Fig. 2, human monoclonal anti-Ebola virus NP (1:2000) and mouse monoclonal anti-FLAG (1:100) antibody were used as the primary antibodies, goat anti-human Alexa-568 (1:1000)

and goat anti-mouse Alexa-647 (1:500) were used as the secondary antibodies.

For samples shown in Fig. 3, human monoclonal anti-NP (1:2000) and goat anti-human Alexa-568 (1:1000) were used.

For samples shown in Fig. 4, rabbit monoclonal anti-V5 (1:500) and mouse monoclonal anti-Ebola VP24 (1:1000, a kind gift from Dr. Yoshihiro Kawaoka) were used as the primary antibodies, goat anti-Rabbit Alexa-647 (1:1000) and goat anti-mouse Alexa-488 (1:500) were used as the secondary antibodies. In addition, trans-complemented APEX2 was stained with Amplex UltraRed at 100 nM (with 0.02% $H_2O_2$ in DPBS) on ice for 20 min prior to fixation. Unreacted Amplex Ultra-Red was washed off with DPBS.

Single-plane confocal images or confocal Z-stacks (interval = 0.224 μm) were acquired with the ZEISS laser scanning microscope (LSM880)-Airyscan system under the superresolution mode, using a plan-apochromat 20×/0.8 numerical aperture M27 objective or an alpha plan-apochromat 63×/1.46 numerical aperture oil Korr M27 objective. All images or stacks that are shown or quantified were Airyscan processed using the Zen black (ZEISS) build-in function.

For Supplementary Fig. 1C, quantification and counting of reconstituted Ebola viral factories (VFs) in single-plane confocal images using ImageJ/Fiji. HEK 293T cells were prepared using identical transfection conditions described in FRAP experiment. VFs were identified on the basis of mNG-fluorescence and analyzed for the count and size distribution using the analyze particle function.

For Fig. 2f, 3D segmentation of Ebola viral factories in confocal Z-stacks was performed in Imaris 9.9.1 using the surface function and automatically thresholding the GFP intensity; segmentation of FLAG-tagged Ebola virus L-foci inside each viral factory was performed using the spot function and automatically thresholding the fluorescence intensity of FLAG-L with an estimated diameter for each spot as 0.5 μm, which generates specific values of distance to nearest neighbor for every random pair of segmented spot.

## Virus infection with EBOV-GFP-ΔVP30

Vero-VP30 cells ($4 \times 10^4$ cells/well) were seeded in IBIDI μ-slides (8 wells high-precision glass bottom). Twenty-four hours later, the monolayer was incubated with EBOV-ΔVP30-GFP (MOI = 3 foci-forming unit/cell) on ice. After 1 h, the monolayer was washed three times with cold DPBS to remove unbound virions and was moved to the 37 °C incubator. Infection was terminated after 16 h. Infected cells were inactivated with 4% PFA for 15 min.

## Thin-section TEM with DAB staining

Sample preparation was adapted from ref. 81. Mattek 35 mm dish with NO#1.5 gridded glass bottom (P35G-1.5-14-C-GRD) was treated with human fibronectin (50 mg/mL) for 1 h at 37 °C incubator, prior to HEK 293T cells seeding ($1 \times 10^5$ cells/well). Twenty-four hours later, the monolayer was either untreated or transfected with plasmids in two different combinations: (1) EBOV Pol1-MG system containing VP35-V5 and L-WT; (2) EBOV Pol1-MG system containing sAPEX2 tagged L-VP35.

For the first combination, 250 ng of pCEZ-NP, 187.5 ng of pCEZ-VP30, 250 ng of pCEZ-VP35-V5, 156.25 ng of pHH21-3E5E-fluc, and 1875 ng of pCEZ-L were used in the co-transfection mix for each 35 mm dish. For the second combination, 250 ng of pCEZ-NP, 187.5 ng of pCEZ-VP30, 250 ng of pCEZ-VP35-V5-sAP, 156.25 ng of pHH21-3E5E-fluc, and 1875 ng of pCEZ-L-sEX were used in the co-transfection mix for each 35 mm dish.

Forty-eight hours post-transfection, cells were fixed first in 2.5% glutaraldehyde-0.1 M cacodylate buffer pH 7.4 + 2 mM $CaCl_2$ at room temperature for 1 min, then fixed in pre-chilled 2.5% glutaraldehyde-0.1 M cacodylate buffer pH 7.4 + 2 mM $CaCl_2$ on ice for 1 h, and washed three times with cold 0.1 M cacodylate buffer pH 7.4 (cacodylate buffer). Unreacted fixative in samples was quenched with cold 20 mM Glycine solution for 5 min on ice, followed by three washes with

cold cacodylate buffer. Samples were stained with 2.5 mM DAB (Sigma #D8001)−0.1 M cacodylate solution in the presence of 1/1000 V of 30% $H_2O_2$ for 45 min on ice and washed three times with cold cacodylate buffer.

Brown DAB stain in samples was confirmed under light microscopy. Samples were then stained in 1% osmium + 1.5% potassium Ferrocyanide in 0.1 M cacodylate buffer on ice for 1 h and washed three times with cacodylate buffer. Samples were dehydrated with increasing concentrations of ethanol (20%, 50%, 70%, 90%, 100%, 100%, 3 min each) and washed once in room temperature anhydrous ethanol. Samples were then infiltrated with Durcupan ACM resin (Electron Microscopy Sciences) using a mixture of anhydrous ethanol: resin (1 V:1 V) for 30 min, then with 100% resin overnight, followed by 48 h polymerization step at 60 °C. Ultrathin sections with a 70 mm thickness of embedded specimen were mounted on 200 mesh hexagonal copper grids with no post-staining step. Electron micrographs were collected using a FEI Tecnai Spirit transmission electron microscope operating at 120 kV with the SerialEM 3.6.12. software.

### Electron tomography and data processing
Electron tomography was performed using a 300 kV Titan Halo equipped with an 8k x 8k direct detector (DE64, Direct Electron). During the procedure, semi-thick (-250 nm) sections of the resin-embedded specimen were imaged at different orientations using a 4-tilt acquisition scheme as described[82], for which the specimen was tilted from −60 to +60 degrees every 0.25 degree at four different azimuthal orientation. For aligning the micrographs, 5 nm colloidal gold particles were deposited on each side of the sections to serve as fiducial markers. The TEM magnification was set at 11,000×, corresponding to a raw pixel size of 0.36 nm. Tomograms were generated with an iterative reconstruction procedure[82], and was binned by a factor of 4 for display and for volume segmentation using Amira 2020 3.1. software. Membrane organelles were segmented manually combined with automatic thresholding. Ribosomes were segmented manually. EBOV viral factories and sAPEX2-tagged EBOV polymerase were segmented with a combination of automatic thresholding and the TopHat tool within the Amira 2020 3.1. software.

### EBOV minigenome assays
Transfection and activity quantification with Pol1-based monocistronic EBOV minigenome system was adapted from ref. 35 and described previously[80]. Transfection with the T7 pol-based EBOV bicistronic minigenome system was adapted from ref. 34. Activity of bicistronic minigenome system was measured in *Renilla* luciferase assay. For each transfection condition, the same cell lysates used in luciferase assay, were pooled from triplicated assay wells and analyzed in western blot.

### Coimmunoprecipitation (coIP)
Co-IP reactions were performed as previously described[80]. HEK 293T cells ($1 \times 10^6$ cells/well) were seeded in 6-wells plates. Twenty-four hours later, cells in each well were transfected with 500 ng of pCMV-mNG-HA or pCAGGS-HA-VP35 or pCAGGS-mNG-HA-VP35 plasmid combined with 500 ng of pCAGGS-NP, with and without 1 μg of pCAGGS-L, using TransIT-LT1 transfection reagent. HA-affinity matrix was used to pull-down protein complexes containing HA-tagged proteins. Equal amount of total protein (1 mg) was used in each co-IP reaction.

### Statistical analysis
Data shown in Figs. 2c, 4c and Supplementary Fig. 1a were calculated and plotted in mean with SD. Statistical analysis was analyzed by two-tailed unpaired t-test with Welch's correction. *P* values were indicated in figure legends. Data shown in Fig. 1f were analyzed using an extra sum-of-squares F test to compare whether the best-fit values of *%Fast*, $k_{fast}$, and $k_{slow}$ differ between two datasets. The resulting *p* value indicating the statistical significance of the difference was reported in the figure.

### Reporting summary
Further information on research design is available in the Nature Portfolio Reporting Summary linked to this article.

## Data availability
All data supporting the findings of this study are available within the paper and its supplementary information. Source data are provided with this paper.

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

## Acknowledgements
We thank Dr. Yoshihiro Kawaoka (University of Wisconsin) for providing the Pol1-based Ebola minigenome system, sharing the anti-VP30 and anti-VP24 antibodies and the EBOV-GFP-ΔVP30 system. We thank Dr. Reika Watanabe (LJI) for providing the EBOV-GFP-ΔVP30 virus stock. We thank Dr. Thomas Hoenen (Friedrich-Loeffler-Institute) for sharing the Ebola tetracistronic trVLP system. We thank Diptiben Parekh (LJI) for plasmid preparations, Dr. Sharon Schendel (LJI) for manuscript editing, NIH S10OD021831 for sponsoring the Zeiss LSM 880 microscope at the LJI microscopy core facility and an Imaging Scientist grant (2019-198153) from the Chan Zuckerberg Initiative to S.M. (LJI). We thank NIH R24GM137200 and S10OD021784 to M.H.E. (UCSD) in support of the National Center for Microscopy and Imaging Research. This research was supported by institutional funds of La Jolla Institute for Immunology to E.O.S. (LJI). J.F. was supported by the Donald E. and Delia B. Baxter Foundation Fellowship.

## Author contributions
J.F. conceived the study and wrote the paper under the supervision of E.O.S.; J.F., G.C., and S.M. designed the experiments, J.F., G.C., S.P., S.M., C.H., and A.A. performed the experiments; J.F. and G.C. analyzed the data, M.H.E. supervised the data acquisition and interpretation for electron microscopy analysis, A.A.D. contributed to data interpretation of intracellular condensates. All authors edited and approved the paper.

## Competing interests
The authors declare no competing interests.
