## [Peer review file · Nature Communications]

REVIEWER COMMENTS

Reviewer #1 (Remarks to the Author):

The viral factories of several Mononegavirales belonging to the Paramyxoviridae, Pneumoviridae and Rhabdoviridae families have been shown to have liquid-like properties. Several characteristics of Ebola virus (EBOV) or Marburg virus factories suggested that this is also the case for Filoviridae. However, this was not formally demonstrated.

In their manuscript, Fang and colleagues have characterized the properties of intracellularly reconstituted Ebola virus (EBOV) factories. This reconstitution allowed them to work in BSL2 facilities (instead of the BSL4 facilities required when working with the virus).

The main results of their article are as follows:

- 1) Using fluorescent and tagged versions of VP35 (HA-mNG-VP35), they show that HA-mNG-VP35 expressed alone formed cytoplasmic gel-like condensates in the cytoplasm whereas coexpression HA-mNG-VP35 and NP yielded condensates that appeared to have liquid properties based on the observation of fusion events, reversible deformability and FRAP experiments.
- 2) Addition of L in the system resulted in the formation of condensates containing all three proteins with a small decrease of the mobile fraction of VP35 inside the condensates.
- 3) A FLAGged version of L reveals that L had a dotted distribution inside the condensates and therefore was not homogeneously distributed.
- 4) Using an EBOV minigenome competent or not for replication, they observe that the L foci inside the condensates were more closely spaced using the minigenome that is deficient for replication.
- 5) They also observe two kinds of morphology for the EBOV viral factories: a typical droplet-like morphology and a more granular, network-like morphology.
- 6) Using a split APEX2-tag they analyze the L-VP35 complex localization by thin section transmission electron microscopy and electron tomography.

To date, this study constitutes the most complete characterization of the viral factories formed by a filovirus. It confirms the observations made on other Mononegavirales and, to my knowledge, presents for the first time data on the localization of the polymerase of a virus of this order.

Nevertheless, it seems that several assertions of the authors could be modulated. In particular, some of their technical approaches are indeed not devoid of artifacts which are now well known in the field.

Major points:

1) Fixation is known to induce major artifacts on protein localization for several biocondensates. The techniques that are used for cell fixation were developed before the identification of liquid organelles and are probably not adapted to their optimal characterization.

This has been exemplified in the article of Irgen-Giorgio et al. (2022) eLife 11:e79903.

<https://doi.org/10.7554/eLife.79903>. Particularly, in this article, the use of glycine to quench the PFA (or the glutaraldehyde) fixation is found to be problematic by further increasing fixation artifacts. Of note, these artifacts are not systematic and some condensates are not sensitive to fixation.

The use of antibodies to visualize the condensates lead also to the appearance structures which are artifactual. This is the case for the ring-like structures which are observed when NP is visualized using an anti-NP antibody and not observed when NP is visualized using NeonGreen fluorescence (in the latter case they are present throughout the VF). This raises some question about the apparent network-like localization of L inside the VF. Indeed in figure 2d, there are two VF and the localization of FLAG-L is completely different in the two cases.

Similarly, the nanoscale localization of split-APEX2 tagged EBOV polymerase complex may be partly artifactual.

I am not saying that the authors are wrong. Nevertheless, they could tone down some of their conclusions a bit and/or try to assess the impact of their fixation protocol and antibodies using fluorescent versions of the proteins and live imaging. They can at least write a paragraph in the discussion indicating the limits of these approaches and the potential artifacts they induce.

2) Why does the GFP concentrate in viral factories (Figure 2D, 2E, 3)? As it does not seem that the viral mRNA translation takes place inside the VF, this raises some questions. Is it also the case in non fixed-cells? This can be checked by live imaging.

3) Using their minimal system, the authors could have determined domains of NP and VP35 (in particular the predicted disordered regions) that are necessary for condensates formation and their liquid properties.

4) line 328: the authors wrote « Surrounding the electron-dense VFs were wire-like fragments that frequently associated with neighboring VFs (Figure 5c, d). » Those wire-like fragments are not really visible.

5) In figure 3b, the authors show distinct pattern of N fluorescence in viral factories (either droplet-like or network-like). Could this reflect a different stage of the infectious cycle? Distinct condensate properties have been observed for Measles Virus (ref 11) and for rabies virus (ref 15) at later stages of infection, this could also be the case for EBOV. The authors only present data obtained at 18h post-infection. At this time 77% of the cells has droplet-like VF. What is the situation at 12h and 24h post-infection?

Minor points:

1) line 32. The authors wrote: « by inducing formation of nonequilibrium, membrane-less compartments ». Why using « non equilibrium », this is not a specificity of these compartments. Life is a non-equilibrium process.

2) Similarly, in line 58, the author wrote: This phase separation in biological systems is ... « based on the intrinsic properties of the protein and RNA constituents and is further influenced by other features of the system's free energy landscape which include kinetic effects such as the macroscopic segregation of dynamically asymmetric mixtures. »

This is a bit vague. They should mention the importance of intrinsically disordered domains in these processes.

3) Captions of Figure 1a and 1b have to be exchanged.

4) In figure 4, panel 4c should be 4b, panel 4d should be 4c, and panel 4b should be 4d.

5) In supplementary figure 3b, the second black arrow from the left of the micrograph does not point to an electron dense viral factory.

Reviewer #2 (Remarks to the Author):

This current manuscript investigated mechanistic details of how Ebola virus (EBOV) VFs orchestrate viral RNA synthesis and defined the organization of the polymerase complex within VFs. The authors took a bottom-up reconstitution approach within cells and deciphered that EBOV VFs are biomolecular condensates. The authors revealed that the EBOV VFs have two distinct morphologies - droplet-like and network-like structures. The internal dynamics of these condensates reduce upon recruitment of the polymerase. Overall, using hybrid imaging approaches, the authors provided key insights into the assembly steps of EBOV particles inside VFs. The insights obtained in this work will contribute new knowledge in the field of viral replication, emerging areas of biomolecular condensate formation and Ebola virus infection mechanism. I feel this work fits the audience of Nat Comms. However, there are a number of issues that I would like the authors to address:

1. Fig. 1a: Where is the loading control? A general tubulin/actin staining would be nice. In the first 'input lane', why the mNG-HA, HA-HP35 and HA-mNG-VP35 conc are variable? This legend needs further clarity.

2. Fig. 1b: discuss what Q-/P-/A-rich regions mean in the text and mention them in the legend.
3. In Fig. 2a, NP is primarily localized at the periphery. Discuss very briefly why this is so other than providing ref.
4. Fig. SI 1C: Is this area or diameter? (line 131)
5. Line 222: Introduce VP30 before
6. Fig. 3: The authors should investigate how dynamic the network-like condensates are using FRAP experiments and also assess L localization. This will reveal and compare the properties of the droplet-like condensates to that of the network-like.
7. Fig. 5/6: Under TEM, most condensates look network-like, and very few are droplet-like. Also, their sizes are mostly below 500 nm. Actually, none of them is more than 2 μm , opposite to what is claimed using fLM (line 131).
8. Fig. 6B: Segmentation (and perhaps colours) needs to improve. The left and right don't match well. See at ~7:40 position, 4:20 position. Also, provide movies of segmented volumes to provide depth information.
9. For various experiments, reconstituted systems were used with minimal components. Therefore, the results are indicative only. The discussion should reflect that.
10. In a few cases, sentences are too long and complex, rewrite those, e.g., Lines: 59-62
11. Fig. 1a/b legend: mislabelled.
12. Line: 458: A schematic model would be a nice addition.
13. What was the electron dose for each tilt series?

Reviewer #3 (Remarks to the Author):

In this work, Fang et al., used transfection-based reconstitution of viral factories of Ebola virus (EBOV) to study the architecture, spatial distribution within the cell, and dynamics of the factories. The authors used various expression constructs for detailed and extensive microscopy analyses of the intracellular condensates, using both confocal and electron microscopy. Most of the analyses were performed using fluorescence recovery after photobleaching (FRAP) assay to study the mobility and diffusion of condensates that are formed by combinations of the VP35, nucleoprotein (NP) and the viral polymerase. In addition, the authors analyzed the localization of the viral polymerase within/outside the condensates using TEM and electron tomography to shed light on how the EBOV factories spatially regulate the synthesis of the viral RNA. Altogether, this is a detailed report on the phase separation properties of EBOV factories. The findings from this report provide insights on the organization of VFs and their

material properties, as well as on the viral polymerase localization within the cell. These insights can facilitate the understanding of the biological role of viral factories in EBOV life cycle and thus may be of interest to the virology and LLPS communities. Thus, I recommend on publication of the work in Nature Communications after addressing the following comments which might help clarifying a few points.

1. There is very limited background information about EBOV viral factories – what is currently known about their building blocks and mechanism of formation according to previous reports? The authors should elaborate on this a bit more in the introduction. This is critical to estimate the biological relevancy of the reconstituted condensates of the current study, as it is not clear how well the building blocks of the reconstituted viral factories used in this study represent the natural EBOV factories based on what is known to date.
2. Can the authors explain why only VP35 was analyzed using FRAP and not NP? It is not clear if this selection is due to labeling limitation or a conceptual decision.
3. I am wondering about the gel-like assemblies formed by HA-mNG-VP35 which was expressed alone. Can the authors comment on how this gel is formed?
4. Page 3 line 150: it is not clear how these VF differ from ones described in line 131.
5. What is the molecular basis of VF formation by HA-mNG-VP35? And by VP35 alone? The authors suggested that the faster recovery of VP35-NP condensates is a result of interactions between the proteins. Please elaborate on the possible modes of interaction between the two proteins vs. that of VP35 alone. i.e., the authors can relate to the pI of each protein and the content and composition of LCDs. Similarly, the authors should suggest what are the interaction modes of the ternary condensates, and what how L immobilizes VP35 inside the condensates.
6. In Section 2 (page 4 line 176) the authors state that their FRAP results show composition-dependent viscoelastic behaviors in live cells. However, I am not convinced that the FRAP analysis and the differences in $t_{1/2}$ values provide information on the viscoelastic properties of the condensates. A more informative and quantitatively accurate technique to analyze viscoelastic properties of biomolecular condensates is microrheology. If the authors are unable to provide microrheology analysis of the binary/ternary condensates, I recommend on rephrasing this statement throughout the text.
7. The network-like behavior of the condensates is interesting. Can the authors suggest what biological functionality this VF network can possibly serve? Was it previously reported for factories of other viruses? The authors should discuss this.

Minor comments:

- The authors should avoid using abbreviations as much as possible and define all abbreviations. For instance, HA is not defined in the text. This is especially important for readers outside the virology field.
- Figure 1: it seems like the text of 1a describes 1b and vice versa.
- Figure 1: are the schematics on the right rely on previous research?

- Results section 2: technical details of the FRAP analysis can be summarized in the methods sections rather than in the main text and figure caption.
- Figure 1: is the reason why the FRAP analysis of VP35+NP+L was performed on n=6 for rather than n=9 as for the other condensates due to difficulties in obtaining VFs of the ternary system? Can the authors please explain this?

REVIEWER COMMENTS

Reviewer #1 (Remarks to the Author):

The viral factories of several Mononegavirales belonging to the Paramyxoviridae, Pneumoviridae and Rhaboviridae families have been shown to have liquid-like properties. Several characteristics of Ebola virus (EBOV) or Marburg virus factories suggested that this is also the case for Filoviridae. However, this was not formally demonstrated.

In their manuscript, Fang and colleagues have characterized the properties of intracellularly reconstituted Ebola virus (EBOV) factories. This reconstitution allowed them to work in BSL2 facilities (instead of the BSL4 facilities required when working with the virus).

The main results of their article are as follows:

- 1) Using fluorescent and tagged versions of VP35 (HA-mNG-VP35), they show that HA-mNG-VP35 expressed alone formed cytoplasmic gel-like condensates in the cytoplasm whereas coexpression HA-mNG-VP35 and NP yielded condensates that appeared to have liquid properties based on the observation of fusion events, reversible deformability and FRAP experiments.
- 2) Addition of L in the system resulted in the formation condensates containing all three proteins with a small decrease of the mobile fraction of VP35 inside the condensates.
- 3) A FLAGged version of L reveals that L had a dotted distribution inside the condensates and therefore was not homogeneously distributed.
- 4) Using an EBOV minigenome competent or not for replication, they observe that the L foci inside the condensates were more closely spaced using the minigenome that is deficient for replication.
- 5) They also observe two kind of morphology for the EBOV viral factories: a typical droplet-like morphology and a more granular, network-like morphology.
- 6) Using a split APEX2-tag they analyze the L-VP35 complex localization by thin section transmission electron microscopy and electron tomography.

To date, this study constitutes the most complete characterization of the viral factories formed by a filovirus. It confirms the observations made on other Mononegavirales and, to my knowledge, presents for the first time data on the localization of the polymerase of a virus of this order.

Nevertheless, it seems that several assertions of the authors could be modulated. In particular, some of their technical approaches are indeed not devoid of artifacts which are now well known in the field.

Major points:

- 1) Fixation is known to induce major artifacts on protein localization for several biocondensates. The techniques that are used for cell fixation were developed before the identification of liquid organelles and are probably not adapted to their optimal characterization. This has been exemplified in the article of Irgen-Giorgio et al. (2022) eLife 11:e79903. <https://doi.org/10.7554/eLife.79903>. Particularly, in this article, the use of glycine to quench the PFA (or the glutaraldehyde) fixation is found to be problematic by further increasing fixation artifacts. Of note, these artifacts are not systematic and some condensates are not sensitive to fixation.

The use of antibodies to visualize the condensates lead also to the appearance structures which are artifactual. This is the case for the ring-like structures which are observed when NP is visualized using an anti-NP antibody and not observed when NP is visualized using NeonGreen fluorescence (in the latter case they are present throughout the VF). This raises some question about the apparent network-like localization of L inside the VF. Indeed in figure 2d, there are two VF and the localization of FLAG-L is completely different in the two cases.

Similarly, the nanoscale localization of split-APEX2 tagged EBOV polymerase complex may be partly artifactual.

I am not saying that the authors are wrong. Nevertheless, they could tone down some of their conclusions a bit and/or try to assess the impact of their fixation protocol and antibodies using fluorescent versions of the proteins and live imaging. They can at least write a paragraph in the discussion indicating the limits of these approaches and the potential artifacts they induce.

Response: We agree with the reviewer and the cited paper (*Irgen-Giorgio et al. 2022, eLife*) that chemical fixation can induce artifacts on the appearance of intracellular LLPS. In our response to the 2) major point, we provided an additional explanation.

We agree with the reviewer that using the anti-NP antibody in immunofluorescence microscopy can lead to biased staining of NP on the periphery of the viral factory. The same applies to the RSV N protein (*Rincheval et al. 2017, Nat. Comm. PMID: 28916773*). We have commented on our system's possible biased antibody staining of EBOV NP (lines #180-182). We fused the mNeoGreen (mNG) tag to the EBOV VP35 protein instead of EBOV NP. Therefore, we only have the NP localization information based on immunofluorescence staining in chemically fixed cells. We did not generate any fluorescence-tagged NP since the incorporation of fluorescence protein in the EBOV NP can reduce the activity of NP by > 90% (*Bodmer & Hoenen, 2022, Viruses; PMID: 35632785*).

We acknowledge that in Figure 2d, there are two viral factories (VFs), one in the reporter-positive cell and one in the reporter-negative cell. Indeed, the localization patterns of FLAG-L inside these two VFs are different. However, we only consider the localization of FLAG-L inside the reporter-positive cell relevant because expression of the minigenome reporter can indicate the active reconstitution of Ebola L-mediated RNA synthesis. In reporter-positive cells, all components of vRNP (NP, VP35, VP30, L, minigenome) are expressed with a stoichiometry that enables replication and transcription of the minigenome. In reporter-negative cells, any of the vRNP components could be missing or at a sub-optimal ratio, likely contributing to the different localization patterns of FLAG-L observed in Figure 2d.

We agree with the reviewer and have added the requested discussion on the technical limitation of our work (lines# 475-489).

2) Why does the GFP concentrate in viral factories (Figure 2D, 2E, 3)? As it does not seem that the viral mRNA translation takes place inside the VF, this raises some questions. Is it also the case in non fixed-cells? This can be checked by live imaging.

Response: We noticed that GFP concentrating in viral factories is an effect of chemical fixation, in line with findings described in *Irgen-Giorgio et al. 2022*. We found that GFP, the reporter protein expressed from the minigenome mRNA, was evenly diffused throughout the entire cell body in live cells. However, upon chemical fixation, the GFP reporter was preferentially retained in membraneless compartments such as the EBOV viral factories and the nucleolus (see Figure 2D and Figure 3A as examples). We commented on this in our revised manuscript (lines# 199-201).

3) Using their minimal system, the authors could have determined domains of NP and VP35 (in particular the predicted disordered regions) that are necessary for condensates formation and their liquid properties.

Response: We thank the reviewer for their insightful suggestions. We agree that the predicted disordered regions in NP and VP35 may play a role in the phase behavior of the EBOV viral factory. We want to point out that *Miyake et al. JVI. 2020 (PMID: 32493824)* has already determined that a region in the EBOV NP (481-500 aa) is necessary for condensates formation in a VP35-dependent manner. The same paper also characterized the propensity of condensate recruitment for a panel of EBOV VP35 truncation mutants when co-expressed with NP. Deletion of both VP35 N-terminal disordered region (1-80 aa) or VP35 C-terminal folded domain (220-340 aa) reduced the recruitment of VP35 to NP-induced condensates. Deletion of VP35 N-terminal disordered region also reduced VP35 protein expression.

In our revised manuscript, we determined that NP C-terminal disordered region (500-739 aa) is dispensable for the liquid properties of the two-component Ebola viral factories (see Supplementary Figure 2d in the revised manuscript).

4) line 328: the authors wrote « Surrounding the electron-dense VFs were wire-like fragments that frequently associated with neighboring VFs (Figure 5c, d). » Those wire-like fragments are not really visible.

Response: We agree with the reviewer that wire-like fragments are not apparent in Figure 5c, d (thin-section TEM images). Instead, these wire-like coils are more evident in our electron tomogram (Figure 6d, e). We removed the statement “Surrounding the electron-dense VFs were wire-like fragments that frequently associated with neighboring VFs” from our revised manuscript and the wire-like drawings from Figure 5e.

5) In figure 3b, the authors show distinct pattern of N fluorescence in viral factories (either droplet-like or network-like). Could this reflect a different stage of the infectious cycle? Distinct condensate properties have been observed for Measles Virus (ref 11) and for rabies virus (ref 15) at later stages of infection, this could also be the case for EBOV. The authors only present data obtained at 18h post-infection. At this time 77% of the cells has droplet-like VF. What is the situation at 12h and 24h post-infection?

Response: We agree that in Figure 3b, the distinct pattern of EBOV NP fluorescence in viral factories may entail a different functional role in the virus infection. We performed additional immunofluorescence analysis and concluded that network-like VFs are not the immediate source of viral nucleocapsid assembly (Figure 3c, d, e; lines #237-250). We discussed possible explanations for the formation and functional role of network-like VFs in our revised discussion (lines #443- 449).

The reviewer compared our result in Figure 3b to the morphological characterization of viral factories in *Zhou et al. 2019, JVI (Measles virus)* and *Nikolic et al. 2018, Nat Comm (Rabies virus)*. This may not be a fair comparison. Both references reported a longitudinal profile of viral factories growing in size as the infection progresses. In our manuscript, two distinct morphologies of Ebola viral factories, not just differences in size, were reported for the same time point post-infection. Similarly, *Nikolic et al. 2018, Nat Comm.* observed some small puncta structures in addition to the Rabies virus Negri bodies. These puncta structures are thought to be condensed Rabies vRNPs. However, the condensed vRNPs (nucleocapsids) are filamentous

and about 1 μm long for the Ebola virus. They are absent nearby network-like viral factories (see Figure 3c, d, e).

We intentionally chose the 18 hours post-infection time point for the morphology analysis because, at this time point, most viral factories are $> 5 \mu\text{m}$ in diameter. We have been focusing on these sizable viral factories in our manuscript. We avoided the 24 hours post-infection because the viral nucleocapsids are disseminated all most everywhere inside cells at this time point (see *Nanbo et al. 2013 Sci Rep. PMID: 23383374*), which could interfere with the determination of droplet-like vs. network-like viral factories. Whereas, Ebola viral factories are too small in EBOV-GFP- ΔVP30 infected cells at 12 hours post-infection (see figure below, scale bar = 10 μm), hence not suitable to characterize the spectrum of different morphologies among sizable viral factories. We decided not to include the figure below in our revised manuscript.

Minor points:

1) line 32. The authors wrote: « by inducing formation of nonequilibrium, membrane-less compartments ». Why using « non equilibrium », this is not a specificity of these compartments. Life is a non-equilibrium process.

Response: We agree with the reviewer and removed the phrase “non-equilibrium” from our text.

2) Similarly, in line 58, the author wrote: This phase separation in biological systems is ... « based on the intrinsic properties of the protein and RNA constituents and is further influenced by other features of the system’s free energy landscape which include kinetic effects such as the macroscopic segregation of dynamically asymmetric mixtures. »

This is a bit vague. They should mention the importance of intrinsically disordered domains in these processes.

Response: We agree with the reviewer and we mentioned the importance of disordered regions in these process in our revised manuscript (lines #54-56).

3) Captions of Figure 1a and 1b have to be exchanged.

Response: We thank the reviewer and we corrected the captions of Figure 1a, b.

4) In figure 4, panel 4c should be 4b, panel 4d should be 4c, and panel 4b should be 4d.

Response: We are not sure why these changes are requested by the reviewer. They appear correctly labelled to our eyes.

5) In supplementary figure 3b, the second black arrow from the left of the micrograph does not point to an electron dense viral factory.

Response: We removed the redundant arrow from the Supplementary Figure 3b (now Supplementary Figure 5b).

Reviewer #2 (Remarks to the Author):

This current manuscript investigated mechanistic details of how Ebola virus (EBOV) VFs orchestrate viral RNA synthesis and defined the organization of the polymerase complex within VFs. The authors took a bottom-up reconstitution approach within cells and deciphered that EBOV VFs are biomolecular condensates. The authors revealed that the EBOV VFs have two distinct morphologies - droplet-like and network-like structures. The internal dynamics of these condensates reduce upon recruitment of the polymerase. Overall, using hybrid imaging approaches, the authors provided key insights into the assembly steps of EBOV particles inside VFs. The insights obtained in this work will contribute new knowledge in the field of viral replication, emerging areas of biomolecular condensate formation and Ebola virus infection mechanism. I feel this work fits the audience of Nat Comms. However, there are a number of issues that I would like the authors to address:

1. Fig. 1a: Where is the loading control? A general tubulin/actin staining would be nice. In the first 'input lane', why the mNG-HA, HA-HP35 and HA-mNG-VP35 conc are variable? This legend needs further clarity.

Response: The loading control (tubulin/actin) is typically shown to confirm that all samples contain an equal amount of the total proteins. In our co-immunoprecipitation (co-IP) experiment, we quantified the total protein concentration in individual cell lysate samples using BCA assay. The same quantity of total protein (1 mg) was adjusted and used in each co-IP reaction. Thus, we do not see the need to probe with a loading control for our input in Figure 1a. We have revised the figure legend to make this information clear (line#510).

Although we used the same mass of expression plasmids in transfection, the different expression levels of mNG-HA, HA-VP35, and HA-mNG-VP35 could reflect different intrinsic protein stability in HEK 293T cells. Different expression promoters (CMV promoter for mNG-HA, CAGGS promoter for HA-VP35, and HA-mNG-VP35) could also account for the different expression levels of each protein bait.

2. Fig. 1b: discuss what Q-/P-/A-rich regions mean in the text and mention them in the legend.

Response: The Q/P/A here are single-letter amino acid codes. We added the explanation in the revised Figure legend (line #499-500).

3. In Fig. 2a, NP is primarily localized at the periphery. Discuss very briefly why this is so other than providing ref.

Response: Previous work, such as *Mina Farag et al. 2022, Nat. Comm.* (PMID: 36513655), indicates that the conformation of proteins at the condensate interface between the coexisting dilute phase (cytosolic NP) and the dense phase (NP inside viral factories) can be different. At the periphery of viral factories, NP may adopt a unique protein conformation that allows the monoclonal antibody (KZ51) we used to detect NP better access to its epitope (lines #180-182).

4. Fig. SI 1C: Is this area or diameter? (line 131)

Response: The x-axis in the original Supplementary Figure 1c is the area. Nevertheless, we agree with the reviewer that in line 131 in the previous version of our manuscript, we referred to the diameter of viral factories. Hence, it's better to show in Figure SI 1c the diameter of viral factories. We updated Supplementary Figure 1c in our revised manuscript with a graph showing the distribution of the Feret's diameter of viral factories. Feret's diameter is a built-in particle analysis tool in FIJI to describe the longest distance between any two points along the selection boundary. This parameter can be a proxy to represent the diameter of viral factories.

5. Line 222: Introduce VP30 before

Response: We now briefly introduce VP30 in our introduction (line #72).

6. Fig. 3: The authors should investigate how dynamic the network-like condensates are using FRAP experiments and also assess L localization. This will reveal and compare the properties of the droplet-like condensates to that of the network-like.

Response: We agree with the reviewer that further characterization of the network-like condensates in Figure 3b could be informative. However, neither the FRAP experiment nor L localization determination are feasible with the EBOV-GFP- Δ VP30 system, in which we observe the apparent network-like viral factories.

- FRAP experiments are live-cell imaging-based experiments using a confocal microscope outside the BSL2+ laboratory. Unfortunately, handling EBOV-GFP- Δ VP30 infected cells may only take place inside the BSL2+ laboratory.
- The precise determination of L localization (e.g., in Figure 2) requires the expression of a FLAG-tagged L, but our EBOV-GFP- Δ VP30 virus carries a wild-type L protein. Modification of the EBOV-GFP- Δ VP30 virus will take additional biosafety approval. On the other hand, the polyclonal antibody against wild-type L is unsuitable for immunofluorescence analysis (Figure SI 1d).

Nevertheless, we included additional immunofluorescence analysis with network-like viral factories in our revised manuscript (lines #237-250, Figure 3c, d, e).

7. Fig. 5/6: Under TEM, most condensates look network-like, and very few are droplet-like. Also, their sizes are mostly below 500 nm. Actually, none of them is more than 2 μ m, opposite to what is claimed using fLM (line 131).

Response: We agree with the reviewer that under TEM, most viral factory condensates are not droplet-like and are smaller than the droplet-like condensates in light microscopy analysis. In the result (line #271-283, Figure 4f-e), the split-APEX2 tag we engineered to the Ebola L-VP35 complex has increased the valence of intermolecular interaction and thus led to the network-like viral factory condensates. In contrast, the reconstituted Ebola viral factory condensates we used in fluorescence light microscopy (fLM) do not have Ebola L-VP35 tagged with split-APEX2. We added the limitation of the different tagging approaches we used in the discussion (line #491).

8.Fig. 6B: Segmentation (and perhaps colours) needs to improve. The left and right don't match well. See at ~7:40 position, 4:20 position. Also, provide movies of segmented volumes to provide depth information.

Response: We have improved the segmentation in Figure 6b in our revised manuscript.

- 1) Adjusted the coloring of our segmented volume and surface.
- 2) Provided additional segmentation in multimembrane vesicles in our tomogram.
- 3) Included a supplementary movie showing segmented volumes in our supplementary information.

9.For various experiments, reconstituted systems were used with minimal components. Therefore, the results are indicative only. The discussion should reflect that.

Response: We have incorporated the above-mentioned limitation in the revised discussion (see line #475-485).

10.In a few cases, sentences are too long and complex, rewrite those, e.g., Lines: 59-62

Response: We rewrote the sentence in line 59-62 (now line #54-57).

11.Fig. 1a/b legend: mislabelled.

Response: We thank the reviewer and we corrected the captions of Figure 1a and 1b.

12.Line: 458: A schematic model would be a nice addition.

Response: We will need more experimental evidence (in future work) to create such a schematic model.

13.What was the electron dose for each tilt series?

Response: The total dose for each tilt series was $1000 \text{ e}^-/\text{\AA}^2$, and in our case the subcellular section was embedded in resin not in vitrified ice. Resin can sustain significantly higher dose than vitrified ice.

Reviewer #3 (Remarks to the Author):

In this work, Fang et al., used transfection-based reconstitution of viral factories of Ebola virus (EBOV) to study the architecture, spatial distribution within the cell, and dynamics of the factories. The authors used various expression constructs for detailed and extensive microscopy analyses of the intracellular condensates, using both confocal and electron microscopy. Most of the analyses were performed using fluorescence recovery after

photobleaching (FRAP) assay to study the mobility and diffusion of condensates that are formed by combinations of the VP35, nucleoprotein (NP) and the viral polymerase. In addition, the authors analyzed the localization of the viral polymerase within/outside the condensates using TEM and electron tomography to shed light on how the EBOV factories spatially regulate the synthesis of the viral RNA. Altogether, this is a detailed report on the phase separation properties of EBOV factories. The findings from this report provide insights on the organization of VFs and their material properties, as well as on the viral polymerase localization within the cell. These insights can facilitate the understanding of the biological role of viral factories in EBOV life cycle and thus may be of interest to the virology and LLPS communities. Thus, I recommend on publication of the work in Nature Communications after addressing the following comments which might help clarifying a few points.

1. There is very limited background information about EBOV viral factories – what is currently known about their building blocks and mechanism of formation according to previous reports? The authors should elaborate on this a bit more in the introduction. This is critical to estimate the biological relevancy of the reconstituted condensates of the current study, as it is not clear how well the building blocks of the reconstituted viral factories used in this study represent the natural EBOV factories based on what is known to date.

Response: We thank the reviewer for his/her comment and we have improved our introduction in the revised manuscript (line#68-80).

2. Can the authors explain why only VP35 was analyzed using FRAP and not NP? It is not clear if this selection is due to labeling limitation or a conceptual decision.

Response: We have added an explanation in the revised manuscript (lines # 101-103). The selection of Ebola VP35 over NP for FRAP analysis is due to labeling limitations. We did not have NP fluorescence-tagged because incorporating a green fluorescence protein in the Ebola NP can severely reduce the activity of NP by > 90% (*Bodmer & Hoenen, 2022, Viruses; PMID: 35632785*).

3. I am wondering about the gel-like assemblies formed by HA-mNG-VP35 which was expressed alone. Can the authors comment on how this gel is formed?

Response: Our manuscript does not have experimental evidence to shed light on the nature of VP35-alone condensates. If we were to speculate, two possible driving forces would mediate these gel-like, HA-mNG-VP35-containing assemblies.

First, Ebola VP35 is known to self-oligomerize through its oligomerization domain (83-145aa), also known as the coiled-coil motif, and can form trimeric or tetrameric parallel bundles (*Zinzula et al. 2019, Structure; PMID: 30482729*). The VP35 primarily forms oligomers through hydrophobic and electrostatic interactions in solution (*Chanthamontri, et al. 2019, Biochemistry; PMID:30592210*), suggesting that when expressed in cells, VP35 can rapidly oligomerize and form stable homo-oligomers.

Second, Ebola VP35 is known to bind the phosphodiester backbone and cap the blunt end of double-stranded RNA (*Leung, et al. 2010, NSMB; PMID: 20081868*) through its C-terminal domain (221-340aa), also known as the interferon inhibitory domain (IID). In this dsRNA-binding mode, four Ebola VP35 IID molecules simultaneously bind one dsRNA molecule, mainly through hydrogen bonding. During virus infection, Ebola VP35 coats the virus-derived dsRNA to evade the cellular sensing of innate immunity. Without virus infection, mammalian cells can

accumulate self-derived, endogenous dsRNA (*Chen & Hur 2022, Nat Rev Mol Cell Biol; PMID:34815573*). Thus, Ebola VP35, when expressed in cells alone, may bind and sequester self-derived endogenous dsRNA.

These multivalent interactions likely mediate the formation of gel-like assemblies when HA-mNG-VP35 is expressed alone. We have added the relevant reference in the revised manuscript (lines # 131-135). We also note that NP increases the internal exchange in VFs likely by reducing the strength or multivalency related to the abovementioned interactions (see response to comment 5 below).

4. Page 3 line 150: it is not clear how these VF differ from ones described in line 131.

Response: The two-component viral factories (VFs) mentioned in line #150 (now line #136) are the same as described in line #131 (now line #122) and the same as being analyzed in Supplementary Figure 1c.

5. What is the molecular basis of VF formation by HA-mNG-VP35? And by VP35 alone? The authors suggested that the faster recovery of VP35-NP condensates is a result of interactions between the proteins. Please elaborate on the possible modes of interaction between the two proteins vs. that of VP35 alone. i.e., the authors can relate to the pI of each protein and the content and composition of LCDs. Similarly, the authors should suggest what are the interaction modes of the ternary condensates, and what how L immobilizes VP35 inside the condensates.

Response: Please find our response to comment #3 for the molecular basis of VP35-alone condensates.

Compared to VP35 alone condensates, the additional heterotypic interaction in the VP35-NP condensates can be mediated through the N-terminal disordered region (20-48 aa) in VP35 forming extensive hydrogen bonds and hydrophobic interactions with residues in the well-folded, NP N-terminal domain (*Leung et al. 2015, Cell Rep; PMID:25865894*). Besides, the predicted disordered region (480-500 aa) in NP interacts with the C-terminal domain of VP35 (221-430 aa) (*Miyake et al. 2020, JVI; PMID: 32493824*). Since negatively charged residues (D/E) are highly enriched in NP (480-500 aa) region, it is plausible that this negatively charged region of NP interacts with the basic patch of the VP35-C terminal domain through electrostatic interactions. We added molecular interactions in our revised manuscript (lines #384-389).

The following speculations based on the pI and LCD of NP and VP35 protein do not support additional modes of interaction between EBOV NP and VP35. The isoelectric point (pI) for VP35 and NP are below 7. Thus, both proteins are positively charged in physiological conditions, as typically seen in RNA-binding proteins. The composition of the low complexity domain (LCD) in both VP35 and NP is biased toward Q (glutamine), which can also contribute to modulating condensate material properties (*Wang et al. 2018, Cell; PMID: 29961577*).

Compared to VP35-NP condensates, the additional heterotypic interaction in the VP35-NP-L condensates may include hydrogen bonds and van der Waals contacts between ordered regions in VP35 tetramer in a protomer-specific manner and the N-terminal domain of L (*Yuan et al. 2022, Nature; PMID:36171293*).

We predict that Ebola L protein can form dynamically arrested networks based on homo-oligomerization through the N-terminal structured region (1-450 aa) of L (*Trunschke et al. 2013,*

Virology; PMID: 23582637) and weak electrostatic interaction through a predicted disordered region (1666-1712 aa: NNSDGHIEREQTTTRDPHDGTERNLVLQMSHEIKRTTIPQENTHQG) in the C-terminal of L. The fraction of VP35 molecules that interact with L is therefore being immobilized, consistent with our FRAP results.

6. In Section 2 (page 4 line 176) the authors state that their FRAP results show composition-dependent viscoelastic behaviors in live cells. However, I am not convinced that the FRAP analysis and the differences in t1/2 values provide information on the viscoelastic properties of the condensates. A more informative and quantitatively accurate technique to analyze viscoelastic properties of biomolecular condensates is microrheology. If the authors are unable to provide microrheology analysis of the binary/ternary condensates, I recommend on rephrasing this statement throughout the text.

Response: We agree with the reviewer that the FRAP experiment cannot directly measure the viscoelastic behaviors of biomolecular condensates. Indeed, microrheology analysis, such as optical tweezer or single-particle tracking (*Alshareedah, et al. 2021, Methods Enzymol.*; PMID: 33453924), is not compatible with intracellular condensates in our study. We rephrased our statements on the viscoelastic behavior in the text and changed it to internal exchange dynamics in our revised manuscript.

7. The network-like behavior of the condensates is interesting. Can the authors suggest what biological functionality this VF network can possibly serve? Was it previously reported for factories of other viruses? The authors should discuss this.

Response: We have performed additional immunofluorescence analysis with these network-like viral factories in Ebola virus-infected cells. We found that these network-like viral factories cannot support viral nucleocapsid assembly, which is necessary to produce infectious virion. In this sense, network-like behavior serves a non-productive role in virus infection. The network-like viral factory is similar to those previously observed in Marburg virus-infected cells (Marburg virus is a filovirus related to Ebola) but has not been reported for other viruses. We added the suggested discussion to our revised manuscript (lines # 437-438).

Minor comments:

1) The authors should avoid using abbreviations as much as possible and define all abbreviations. For instance, HA is not defined in the text. This is especially important for readers outside the virology field.

Response: We agree with the reviewer and have now removed several abbreviations (BSL-Biosafety level; RABV-Rabies virus; vRNA-genome; cRNA-antigenome) in our revised manuscript to improve our communication with a broader audience.

2) Figure 1: it seems like the text of 1a describes 1b and vice versa.

Response: We thank the reviewer and we corrected the captions of Figure 1a and 1b.

3) Figure 1: are the schematics on the right rely on previous research?

Response: Yes, in Figure 1c-e, the schematics on the right rely on a collection of previous research that characterized the protein-protein interactions of Ebola virus NP, VP35 and L protein. We have also summarized these molecular interactions in our response to major comment #5.

4) Results section 2: technical details of the FRAP analysis can be summarized in the methods sections rather than in the main text and figure caption.

Response: We moved technical details of the FRAP analysis previously in the results now to the figure caption and methods.

5) Figure 1: is the reason why the FRAP analysis of VP35+NP+L was performed on n=6 for rather than n=9 as for the other condensates due to difficulties in obtaining VFs of the ternary system? Can the authors please explain this?

Response: We thank the reviewer's comment. In the legend in Figure 1, we made a mistake about the number of cells being analyzed in each condition. The N=6 was intended to be VP35 alone condensates. For both VP35+NP and VP35+NP+L, N=9 cells were analyzed. Obtaining sizable VFs in the VP35 alone condition was challenging because most cells do not form a dense phase of VP35. For those few cells with a VP35-dense phase, these VFs are relatively small for a 1 μm -bleach spot size.

REVIEWERS' COMMENTS

Reviewer #1 (Remarks to the Author):

In the revised version of their manuscript, Fang and colleagues took my previous comments into account and clarified several aspects of their work.

I would suggest they cite Irgen-Giorgio et al. (2022) eLife 11:e79903.

<https://doi.org/10.7554/eLife.79903> about fixation and immunofluorescence-induced artifacts in the paragraph that they added in the discussion of the technical limitation of their work (lines# 475-489). This would be fair.

Reviewer #2 (Remarks to the Author):

The authors have clarified all my doubts and revised the MS appropriately. This manuscript is now ready for publication.

Reviewer #3 (Remarks to the Author):

The authors have addressed my concerns, the manuscript has improved, and I now recommend its publication.

REVIEWERS' COMMENTS

Reviewer #1 (Remarks to the Author):

In the revised version of their manuscript, Fang and colleagues took my previous comments into account and clarified several aspects of their work.

I would suggest they cite Irgen-Giorgio et al. (2022) eLife 11:e79903. <https://doi.org/10.7554/eLife.79903> about fixation and immunofluorescence-induced artifacts in the paragraph that they added in the discussion of the technical limitation of their work (lines# 475-489). This would be fair.

Response: We now added the suggested reference to the paragraph discussing technical limitations of our work (line #480).

Reviewer #2 (Remarks to the Author):

The authors have clarified all my doubts and revised the MS appropriately. This manuscript is now ready for publication.

Reviewer #3 (Remarks to the Author):

The authors have addressed my concerns, the manuscript has improved, and I now recommend its publication.